# Anti-Planktonic and Anti-Biofilm Properties of Pentacyclic Triterpenes—Asiatic Acid and Ursolic Acid as Promising Antibacterial Future Pharmaceuticals

**DOI:** 10.3390/biom12010098

**Published:** 2022-01-07

**Authors:** Zuzanna Sycz, Dorota Tichaczek-Goska, Dorota Wojnicz

**Affiliations:** Department of Biology and Medical Parasitology, Wroclaw Medical University, 50-345 Wroclaw, Poland; sycz.zuzanna@gmail.com (Z.S.); dorota.wojnicz@umw.edu.pl (D.W.)

**Keywords:** asiatic acid, ursolic acid, pentacyclic triterpenes, bacterial biofilm, chronic and recurrent infections

## Abstract

Due to the ever-increasing number of multidrug-resistant bacteria, research concerning plant-derived compounds with antimicrobial mechanisms of action has been conducted. Pentacyclic triterpenes, which have a broad spectrum of medicinal properties, are one of such groups. Asiatic acid (AA) and ursolic acid (UA), which belong to this group, exhibit diverse biological activities that include antioxidant, anti-inflammatory, diuretic, and immunostimulatory. Some of these articles usually contain only a short section describing the antibacterial effects of AA or UA. Therefore, our review article aims to provide the reader with a broader understanding of the activity of these acids against pathogenic bacteria. The bacteria in the human body can live in the planktonic form and create a biofilm structure. Therefore, we found it valuable to present the action of AA and UA on both planktonic and biofilm cultures. The article also presents mechanisms of the biological activity of these substances against microorganisms.

## 1. Introduction

In recent years, there has been a growing interest in natural sources of medicinal substances. In the era of the growing resistance of bacteria to antibiotics, an intensive search is being made for effective and safe plant-derived compounds that could become a useful tool both in terms of prevention and treatment of diseases of bacterial aetiology. It is believed that plant products used as part of phytotherapy may support standard pharmacotherapy. Their potential use in terms of controlling chronic and/or recurrent inflammation caused by bacterial biofilm formation is of particular interest.

A bacterial biofilm is a community of intercommunicating microorganisms settled on a specific surface, adhering to each other, and surrounded by a layer of organic and inorganic substances produced by these microorganisms. The biofilm consists of aggregates of bacterial cells embedded within an extracellular polymeric substance (EPS), covering biotic and abiotic surfaces [1]. Biofilm formation proceeds in multiple stages: (1) initial adhesion (reversible adhesion); (2) irreversible adhesion, EPS production; (3) microcolony formation; (4) matrix formation; (5) dispersal [1]. A bacterial biofilm can be formed by cells of one or more microbial species. Multi-species biofilms occur in the natural environment, human body, on biomaterial surfaces, and in industrial installations. They are aetiological agents of chronic, recurrent, and nosocomial infections. Biofilms pose a significant threat to human health and life [1,2,3].

Problems with the treatment of chronic and/or recurrent infections caused by bacteria living in biofilm result not only from the difficulty of penetration of drugs through its structure, but they may also be caused by changes in the expression of genes encoding virulence factors, genes that are important in the process of formation and development of the biofilm structure, and genes encoding drug transporters and regulatory proteins [4,5,6].

Bacterial cells forming a biofilm structure have an altered metabolism compared to forms living in the suspended matter, that is, planktonic forms. In deeper layers of the biofilm, there are super-resistant cells, “persisters”, that have low metabolic activity, which weakens the effect of antimicrobial agents, so that these cells can survive in the presence of lethal concentrations of antibiotics and re-establish the biofilm after the end of the therapeutic cycle [5,7].

The structure of the biofilm and the low metabolic activity of the bacteria living in it mean that the minimum drug concentrations necessary to inhibit their growth are even 100–1000 times higher than the concentrations of the same drugs used to eliminate bacteria that grow as planktonic forms. Such high doses of antibiotics cannot be used in the pharmacotherapy of human infections due to their toxicity [4,5,8]. Bacterial resistance to antibiotics has been a major challenge for public health, especially when associated with biofilm-forming cells that are more resistant to drug treatment. Therefore, in recent years, antimicrobial agents derived from natural sources have attracted great interest due to their potential activity against multidrug-resistant bacteria. Plant metabolites can be such effective antimicrobial compounds that act against planktonic and biofilms forms of pathogens [9,10,11]. Their antibacterial effects on the planktonic forms of bacteria may contribute to a reduced adherence of bacteria to the surface and biofilm formation. The newest research show that the antibacterial activity of AA is attributed to the damage to the bacterial membrane and the enhanced leakage of potassium ions and nucleotides [12]. It also has been demonstrated that UA could decrease the viability and structural integrity of biofilms by occupying the catalytic centre of glucosyltransferases [13] or suppression of *gft* genes’ expression [14] and in consequence leading to inhibition of EPS formation.

For the reasons discussed above, plant-derived compounds are being sought; they may be applied as an alternative or supportive strategy to antibiotic therapy, as well as they may prevent biofilm formation or facilitate biofilm eradication. In recent years, there has been an increasing interest in pentacyclic triterpenes (TPs). They are polycyclic organic compounds of plant origin, used in traditional phytotherapy due to their wide spectrum of valuable medicinal properties. Plants rich in TPs, for example, *Centella asiatica* and *Arctostaphylos uva-ursi*, are frequently used prophylactically in the form of dietary supplements and nutraceuticals, as well as they are used as a support for the treatment of many diseases, especially those with inflammatory causes [15,16,17].

## 2. Asiatic Acid

Asiatic acid (AA; 2α,23-dihydroxyursolic acid) is a monocarboxylic acid derived from the hydride of ursane. In the structural formula, ursane is substituted with a carboxyl group at C-28 and hydroxyl groups at C-2, C-3, and C-23 (stereoisomer 2α, 3β) (Figure 1A). The molecular formula can be presented as C_30_H_48_O_5_, the molecular weight of AA is 488.70 g/mol [18,19]. AA, like other TPs, is a secondary metabolite protecting a plant against insect and microbial attack [20]. Particularly high concentrations of AA in its free form and as an aglycone with attached carbohydrate residues (asiaticoside) can be found in the leaves of *C. asiatica*, known as Gotu kola or kodavan. The total fraction of TPs of *C. asiatica* contains 30% AA, 40% asiaticoside and 30% madecassic acid [21]. This plant can be found in South Africa, Australia, Oceania, and Southeast Asian countries (mainly India and China, but also Japan, Malaysia, and Indonesia) [21]. For over 3000 years, *C. asiatica* has been one of the main medicinal ingredients used in traditional African, Ayurvedic and Chinese medicine as a panacea and it has been recommended primarily for neuropsychiatric disorders and for the treatment of wounds, leprosy, and syphilis [21,22]. Nowadays, in view of the fact that this plant shows an overall health-promoting effect, in its countries of origin it is consumed prophylactically both in the form of nutraceutical preparations and as an addition to salads and drinks. Moreover, *C. asiatica* is an ingredient in ointments, cosmetics, and toothpaste [21,22].

## 3. Ursolic Acid

Ursolic acid (UA; 3β-hydroxy-urs-12-en-28-oic acid) less commonly known as urson, is a hydroxy monocarboxylic acid derived from the hydride of ursane. In the structural formula, ursane is substituted with a carboxyl group at C-28 and a 3β-hydroxyl group at C-3 (Figure 1B). The molecular formula can be presented as C_30_H_48_O_3_, the molecular weight of UA is 456.70 g/mol [23,24,25,26].

UA is most found in the free form or as an aglycone in the case of triterpenoid saponins. UA is present in many plant species belonging to angiosperms, ferns, and bryophytes [27,28,29,30]. Its particularly high concentrations can be found in the leaves of *A.uva-ursi*, also known as bearberry. The leaf of *A. uva-ursi* is a pharmacopoeial raw material and it contains approx. 0.75% UA and several other pharmacologically active compounds: 6–12% arbutin and its metabolites (galoylarbutin, methylarbutin, free hydroquinone), approx. 20% tannins, 1.2–1.5% flavonoids (quercetin, isoquercetin, myricetin, kaempferol, hyperoside), uvaol, polyphenolic acids (6% gallic acid, syringic acid, p-coumaric acid, ellagic acid, and quinic acid) [31,32].

*A. uva-ursi* can be found in the arctic and temperate zones of the northern hemisphere—in Europe, Asia, and North America. In Poland, it can be found mainly in the northern part of the lowlands, in dry pine forests and heathlands, rarely in the mountains. In 2014, *A. uva-ursi* was included in the list of species under strict protection (Red List) [33]. Leaves of *A. uva-ursi* have been used for centuries in European traditional medicine as a diuretic, astringent, and antiseptic to treat urinary tract infections (UTIs) and relieve pain concerning nephrolithiasis. In contrast, Native Americans used leaves of *A. uva-ursi* to treat headaches, prevent and treat scurvy and prevent miscarriages. In modern phytotherapy, the leaf of *A. uva-ursi* is used alone in the form of decoctions and as an ingredient of urological herbal mixtures. Moreover, the *A. uva-ursi* leaf extract is an ingredient of pharmaceutical compounded preparations—liquids and pills with antibacterial, anti-inflammatory, and diuretic properties, which are recommended for the treatment of UTIs [34]. UA is also one of the biologically active triterpenoids found in several species of bryophytes: *Sphagnum magellanicum*, *Sphagnum tenellum*, *Sphagnum rubellum*, *Polytrichum juniperum*, *Sphagnum fallax*, and *Aulacomnium palustre*. However, only the extract from *S. magellanicum* showed antimicrobial activity [27].

## 4. Pharmacological Activities of AA and UA

Many studies have proved several molecular and pharmacological properties that are common to AA and UA: antioxidant, anti-inflammatory, anti-free radical, cytoprotective, apoptosis-regulating, and receptor- and enzyme-modulating activities. In various in vitro and in vivo studies, AA and UA have also been found to affect growth factors, transcription factors, and cell signalling. The above-mentioned properties result in the following therapeutic actions that are common to AA and UA: anticancer, hypotensive, cardioprotective, anti-infarction, anti-stroke, antihyperlipidemic, antidiabetic, hepatoprotective, gastroprotective, nephroprotective, diuretic, immunostimulatory, neuroprotective, nootropic, anti-Parkinson’s, anti-Alzheimer’s, anti-osteoporosis, antiprotozoal (especially antimalarial), antifungal, antimycobacterial, antiviral (anti-HIV, anti-HCV) and radioprotective (anti-aging) actions. Furthermore, AA supports the treatment of burns and non-healing diabetic wounds, and it also exhibits spermicidal activity. UA, in turn, can accelerate wound healing, stimulate muscle growth and reduce fat gain, as well as it is an ingredient of dermatological preparations, cosmetics, and nutritional supplements for athletes [18,19,23,24,26,35,36,37,38,39,40,41,42,43,44].

## 5. The Mechanisms of Antibacterial Activity of AA and UA

Antibacterial activity is one of the many properties of pentacyclic triterpenes. Their activity is related to changes in the structure and functioning of the bacterial cell structures (cell membrane, adhesins), cell morphology, gene expression, and processes such as adhesion and biofilm formation (Figure 2) [10,13,14,45,46,47]. The available papers present the effects of AA and UA against both Gram-positive and Gram-negative bacteria. Both acids showed better antibacterial activity against Gram-positive bacteria. This activity is related to differences in terms of the structure of cell envelopes of these two groups of bacteria. In the envelopes of Gram-negative bacteria, unlike Gram-positive bacteria, there is an outer membrane that can impede the penetration of acids into the bacterial cell [46,48]. Therefore, there are significant differences in terms of the values of AA and UA concentrations that inhibit the growth of bacteria from both groups. Higher MIC values are usually observed for Gram-negative bacteria.

In studies devoted to the antibiofilm effects of AA and UA, there was a more effective effect of AA, probably resulting from the slightly different molecular structure of this compound compared to UA. This is because AA has an additional hydroxyl group at C-23, which gives a more hydrophilic nature to the whole molecule, making it easier for this acid to penetrate inside the EPS of the biofilm and thus reach bacterial cells [19,49].

The anti-adhesion activity of AA and UA, contributing to the reduced ability of bacteria to adhere to epithelial cells (e.g., uroepithelium), is one of the mechanisms of their antimicrobial activity [10]. Impaired adhesion may be associated with morphological changes in bacterial cells and changed expression of genes encoding virulence factors such as P fimbriae, curli fimbriae, and hydrophobicity. AA- and UA-induced morphological changes and reduced bacterial motility impair the colonisation of tissues and abiotic surfaces and thus they impair biofilm formation.

Another known mechanism for the antibacterial activity of AA and UA is the impairment of the integrity and irreversible damage to the bacterial cell membrane. It has been determined that AA impairs potassium cation (K+) homeostasis, which is regulated by Kdp membrane transporters. K+ is involved in many aspects of bacterial physiology, for example, growth, survival, and virulence. AA increases the release of K+ from the cytoplasm and excessive loss of K+ leads to bacterial cell death. Moreover, the treatment of bacteria with AA causes an unfavourable release of DNA and RNA nucleotides, which may also cause cell death. Although AA can damage the lipid membrane, it may not be able to penetrate the EPS barrier that is a component of the biofilm [11,50].

The molecular mechanism of UA activity also involves the inclusion of its molecules into the inner membrane of the bacterial cell and interference with the formation of hydrogen bonds between phosphatidylethanolamine (PE) and phosphatidylglycerol (PG), which are major components of this membrane. UA molecules can also disorganise the packing of hydrophobic chains of phospholipid membranes. It is also possible that UA not only disrupts the PE-PG packing but also locally alters their mutual ratios, making bacteria more susceptible to antibiotics [51].

After a thorough understanding of the mechanism of action of AA and UA, the next issue to be addressed is to improve their low in vivo bioavailability that limits the clinical application of TPs in question. Therefore, structural modifications of UA have been conducted in recent years to improve its biological activity and bioavailability. The aim of one of the studies by Gu et al. [52] was to design and synthesise novel carbazole derivatives of UA and determine their antibacterial activity against Gram-positive (*Bacillus subtilis*, *Staphylococcus aureus*) and Gram-negative (*Escherichia coli*, *Pseudomonas fluorescens*) strains. Most of the synthesised UA derivatives showed inhibitory activities against both Gram-positive and Gram-negative bacteria, adopting minimum inhibitory concentration (MIC) values of 3.9–15.6 mg/L, which were similar to the MIC values of amikacin—the positive control in that study. These results proved that the presence of the carbazole moiety alters the physicochemical properties of UA, including improvement of its water solubility, which leads to improved bioavailability and biological activity compared to the parent UA molecule.

Huang et al. [45] synthesised various UA derivatives containing two additional hydroxyl groups in the A-ring. Most of these derivatives showed a significantly enhanced activity against Gram-positive bacteria (*B. subtilis* ATCC 6051, *S. aureus* ATCC 25923, and *Streptomyces scabies*), with negligible or weak activity against Gram-negative bacteria (*E. coli* ATCC 25922 and *Ralstonia solanacearum* ATCC 11696) compared to UA. The weak activity of UA and its derivatives is due to the presence of an outer membrane in Gram-negative bacteria, which is a selective barrier against the penetration of these compounds into the cell. The structure-activity relationship analysis of UA derivatives showed that the introduction of two hydroxyl groups into UA at C-1 and C-2 along with a short alkyl ester group at C-17 strongly enhances the growth-inhibitory activity of Gram-positive bacteria. The authors showed that UA derivatives alter the expression levels (up- or down-regulation) of genes related to peptidoglycan metabolism and cellular respiration metabolism, as well as genes involved in bacterial virulence.

Usmani et al. [53] investigated the antimicrobial potential of UA and its various amide derivatives against nosocomial pathogen *Acinetobacter baumannii* ATCC 19606 and its clinical colistin-resistant strains. It is interesting that despite the lack of antibacterial properties of pure UA, the authors showed very good antibiofilm properties of one of its derivatives (3-β-N-(2′,4′-dinitrophenyl)-3-hydroxyurs-12-en28-amide) against all tested *A. baumannii* strains. The authors also found that this amide derivative down-regulates the expression of biofilm development (*bap*) and quorum sensing (*abaR*) genes of *A. baumannii* suggesting this compound might hinder quorum sensing leading to stop biofilm formation.

Ghasemzadeh et al. [54] investigated the antibacterial properties of UA as well as UA-loaded chitosan nanoparticles (UA-Ch-NPs). The MIC values of UA and UA-Ch-NPs against *S. aureus* were 64 and 32 mg/L, respectively. The authors demonstrated that UA-Ch-NPs significantly decreased the expression of *icaA* and *icaD* genes, which are engaged in biofilm formation, indicating that UA-Ch-NPs could be a promising material for antibacterial and antibiofilm applications.

The weaker activity of UA compared to AA is due to its poor solubility in water. To improve the bioavailability of UA, Oprean et al. [55] encapsulated its molecules in polyurethane nanostructures, acting as carriers, and then they determined the antimicrobial activity of these complexes against *Bacillus cereus* and *B. subtilis* strains. Unfortunately, contrary to expectations, “encapsulation” reduced the antimicrobial activity of UA. Therefore, further studies are necessary to improve its antimicrobial activity.

However, there are relatively few literature reports that include studies describing the antimicrobial effects of AA and UA. Experiments conducted in recent years have proved that these acids, alone and combined with antibiotics, have significant antimicrobial activity. Existing studies focus primarily on bacteria living in the suspended matter, that is, planktonic forms. MIC and minimum bactericidal concentration (MBC) values of both acids were determined and their effects on bacterial survival, cell morphology, and bacterial membrane structure were described. The effects of AA and UA on the cell surface hydrophobicity, motility, and synthesis of fimbriae, which are virulence factors of significant importance for biofilm formation, were also shown in those studies

## 6. Antibacterial Properties of Pentacyclic Triterpenes

Planktonic forms of bacteria are free-living single cells that move freely in body fluids. They are more susceptible to used antibiotics compared to their biofilm forms, and they are more easily identified and eliminated by the host immune system [56]. In many cases, gene expression differs in terms of planktonic and biofilm forms, which contributes to their morphological and physiological diversity [57]. It should be noted that planktonic forms when released from the biofilm structure, become the source of subsequent outbreaks of infection. Although planktonic forms represent only 0.1% of the total microbial biomass, they also play a key role in bacterial infections [58]. Therefore, articles concerning the effects of AA and UA on bacteria living in planktonic forms were reviewed.

### 6.1. Anti-Planktonic Activity of Asiatic Acid

The studies presented below used AA isolated by researchers from various plants rich in pentacyclic triterpenes (Figure 3) or purchased in pure form. It should also be noted that these articles include a very diverse range of research.

Some articles are limited to identifying only MIC values (Table 1) or zones of growth inhibition (Table 2). Others are extended to studies concerning the effects of this acid on the morphology and virulence traits of analysed bacteria.

The MIC value is the lowest concentration of the antibacterial substance that inhibits bacterial growth. Determination of the MIC is the most basic laboratory measurement of the activity of an antimicrobial agent against microorganisms. A low MIC value indicates that less antibacterial agent concentration is needed to inhibit bacterial growth. Knowing the MIC values of antibiotics is very important as it allows clinicians to select the appropriate antibiotic for a patient with a specific infection and determine the effective dose of the antibiotic.

Bacteria exposed to an insufficient concentration of a particular drug can evolve resistance to this drug. Therefore, the knowledge of MIC values is helpful in the treatment of patients and prevents the development of drug-resistant strains.

The antimicrobial activity of pure compound isolated from crude extract can be considered significant (very good) when its MIC is below 10 mg/L, moderate (good) when between 10 and 100 mg/L, and low (weak) when more than 100 mg/L [59].

The data in Table 1 indicate that AA exhibits a variety of antimicrobial activities. AA isolated from *S. guineense* leaves showed significant activity against Gram-positive *B. subtilis* (MIC = 0.75 mg/L) and Gram-negative *E. coli* (5 mg/L) [60]. AA showed moderate activity against most of the tested bacteria (10 < MIC <100 mg/L) [11,46,50,60,61,62]. Weak AA activity has been described for Gram-positive strains of *E. faecalis* [46,63] and *S. aureus* [63]. Gram-negative strains of *E. coli*, *P. aeruginosa*, and *E. cloacae* as well as Gram-positive *S. aureus* showed the lowest sensitivity to AA (MIC >128 mg/L) [9,63,64]. Among these strains were uropathogenic *E. coli*, for which the AA MIC ranged from 512 mg/L to >1024 mg/L [9,10,47].

**Table 1 biomolecules-12-00098-t001:** MIC values of asiatic acid against Gram-positive and Gram-negative bacteria.

Bacterial Group	Species	Asiatic Acid			Ref.
		MIC [mg/L]	Source	Antibacterial Activity	
Gram-positive	*Bacillus cereus*	36–44	purchased *^B^	good	[50]
	*Bacillus subtilis*	0.75	*S. guineense*	very good	[60]
	*Clostridium difficile*	10–20	purchased *^A^	good	[11]
	*Enterococcus faecalis*	18–22	purchased *^B^	good	[50]
		64–128	purchased *^A^	good–weak	[46]
		128	*S. lancifolia*	weak	[63]
	*Listeria monocytogenes*	32–40	purchased *^B^	good	[50]
	*Staphylococcus aureus*	20–160	*C. asiatica*	good–weak	[61]
		26–30	purchased *^B^	good	[50]
		>128	*S. lancifolia*	weak	[63]
Gram-negative	*Enterobacter cloacae*	1024	purchased *^A^	weak	[9]
	*Escherichia coli*	5	*S. guineense*	very good	[60]
		20–28	purchased *^B^	good	[50]
		>128	*S. lancifolia*	weak	[63]
		512–>1024	purchased *^A^	weak	[10]
		512–>1024	purchased *^A^	weak	[47]
		>1024	purchased *^A^	weak	[9]
	*Pseudomonas aeruginosa*	32–40	purchased *^B^	good	[50]
		64	*S. robusta*	good	[62]
		>128	purchased **^A^	weak	[64]
		>128	*S. lancifolia*	weak	[63]
		>1024	purchased *^A^	weak	[9]
	*Salmonella typhimurium*	30–34	purchased *^B^	good	[50]
	*Shigella sonnei*	30	*S. guineense*	good	[60]

* purchased from Sigma-Aldrich Chemicals Inc.; ** purchased from LKT Laboratories; ^A^—synthetic; ^B^—isolated from *C. asiatica*.

Djoukeng et al. [60] studied the effects of AA isolated from a methanol extract of *S. guineense* leaves. That acid revealed significant antibacterial activity against *B. subtilis*, *E. coli*, and *S. sonnei* strains and its MIC values were 0.75, 5.0, and 30.0 mg/L, respectively. Taemchuay et al. [61] found that both MIC and MBC values, determined against 30 clinical strains of *S. aureus* and the reference strain *S. aureus* ATCC 25923, were in the range of 20–160 mg/L for AA isolated from *C. asiatica* leaves. Acebey-Castellon et al. [63] determined that AA derived from methanolic extract of *S. lancifolia* leaves exhibited stronger antibacterial activity against *E. faecalis* ATCC 29212 (MIC = 128 mg/L) and weaker activity against *S. aureus* ATCC 25923, *E. coli* ATCC 25922, and *P. aeruginosa* ATCC 27853 (MICs > 128 mg/L).

The effect of AA on clinical *E. coli* strains isolated from the urine of patients with pyelonephritis was also analysed by Wojnicz et al. [10,47]. The articles identified the MIC values of AA against the analysed strains, which ranged from 512 mg/L to >1024 mg/L. Moreover, the effects of AA on virulence traits associated with biofilm formation and development by these bacilli were presented. AA at a concentration of 10 mg/L was sufficient enough to cause a loss of haemagglutinating capacity associated with the presence of P fimbriae. On the other hand, AA at a concentration of 40 mg/L inhibited the synthesis of curli fimbriae and significantly reduced the motility of bacteria. AA at a concentration of 50 mg/L revealed anti-haemolytic activity [47] and reduced the hydrophobic nature of bacterial cell surfaces [10]. At a concentration of 40 mg/L, AA also affected the adhesion of *E. coli* to uroepithelial cells, significantly reducing the number of bacilli adhering to the epithelium. AA at a concentration of 50 mg/L also induced morphological changes in bacterial cells. An increased percentage of short filaments (5–15 μm) and the presence of long filaments (>15 μm) were observed in cultures treated with AA [10].

The study by Liu et al. [50] revealed the antimicrobial activity of AA against Gram-negative bacteria: *E. coli* O157:H7, *S. typhimurium* DT104, *P. aeruginosa*, as well as Gram-positive bacteria: *L. monocytogenes*, *S. aureus*, *E. faecalis*, *B. cereus*. The MIC values of that acid against those bacterial strains were in the range of 20–40 mg/L, while the MBC values were in the range of 32–52 mg/L. AA reduced the survival of all bacterial strains as early as in 6-h cultures. AA at 1 × MIC concentration impaired cell membrane integrity in 40–56% bacterial cells, while at 2 × MIC concentration it caused loss of K+ ions and nucleotides in 71–89% of bacterial cells.

Wojnicz et al. [46] studied in detail the effect of AA on Gram-positive clinical strains of *E. faecalis*, the bacteria associated with serious nosocomial UTIs, particularly among catheterised patients. Those strains were resistant to, for example, gentamicin, nitrofurantoin, ampicillin, and trimethoprim/sulphamethoxazole. The MIC values of AA against the analysed strains were 64–128 mg/L. The anti-growth effect of sub-inhibitory concentrations (0.75 × MIC) of AA was observed after 2-, 4-, 6- and 24-h incubation. AA showed the highest activity after 24 h, where bacterial survival was reduced by 250-fold compared to the control sample. It was also observed that *E. faecalis* cells were larger and formed aggregates. These changes may be caused by AA impairing cell division, resulting in changes in cell morphology such as increased cell diameter or the occurrence of aggregates. Furthermore, the authors showed that AA significantly reduced or completely inhibited the ability of *E. faecalis* to synthesise enzymes and toxins that damage host tissues such as lipase, lecithinase, gelatinase, and haemolysins.

Harnvoravongchai et al. [11] found the potent antimicrobial activity of AA against clinical strains of *C. difficile*. The MIC values of AA were 10–20 mg/L. AA inhibited bacterial motility, induced membrane damage and morphological changes in cells, thus causing leakage of intracellular substances (proteins and nucleic acids) as early as 30 min of exposure. AA at concentrations of 10 and 100 mg/L damaged the spore surface, impairing the sporulation and germination of spores. According to the study by Han et al. [39], AA solution at a concentration of 5 µM (10%), being a part of a dressing for the treatment of diabetic non-healing wounds showed comparable antibacterial activity against *E. coli* ATCC 8739 and *S. aureus* ATCC 29213 as standard dressings soaked in penicillin at a concentration of 25 mg/L. In contrast, AA at a concentration of 15 µM (30%) showed a stronger inhibitory effect against both tested strains than penicillin.

In the study conducted by Bharitkar et al. [62], AA isolated from *S. robusta* resin exhibited good antimicrobial activity against *E. coli* ATCC 25938 and *P. aeruginosa* with the MIC value of 64 mg/L.

Some researchers define the antimicrobial activity of AA by determining the zone of bacterial growth inhibition around a disc soaked with a specific concentration of it (Table 2). The size of the growth inhibition zone, given in millimetres, indicates the sensitivity of the bacteria to the test substance. The results are interpreted depending on the diameter of the inhibition zone as follows: the zone <7 mm is considered as inactive, 7–12 mm as active, >12 mm as very active [65].

**Table 2 biomolecules-12-00098-t002:** Growth inhibition zones and corresponding concentration values of asiatic acid against Gram-positive and Gram-negative bacteria.

Bacterial Group	Species	Inhibition Zone [mm]	Asiatic Acid			Ref.
			Concentration [mg/L]	Source	Antibacterial Activity	
Gram-positive	*Bacillus cereus*	10.0	250	*M. malabathricum*	active	[66]
		11.5	500		active	
		12.0	1000		active	
		13.5	2000		very active	
	*Bacillus subtilis*	7.0	25	*C. asiatica*	active	[67]
		9.0	50		active	
		15.0	75		very active	
		17.0	100		very active	
	*Staphylococcus aureus*	8.0	1000	purchased *^C^	active	[68]
		9.0	250	*M. malabathricum*	active	[66]
		10.0	500		active	
		10.5	1000		active	
		11.0	2000		active	
		7.0	25	*C. asiatica*	active	[67]
		8.0	50		active	
		12.0	75		active	
		13.0	100		very active	
	*Streptococcus pneumoniae*	7.0	1000	purchased *^C^	active	[68]
Gram-negative	*Escherichia coli*	7.0	1000	purchased *^C^	active	[68] [62]
		7.0	64	*S. robusta*	active	
	*Helicobacter pylori*	12.0	1000	purchased *^C^	active	[68]
		8.0	500		active	
	*Klebsiella pneumoniae*	8.0	1000	*M. malabathricum*	active	[66]
		9.0	2000		active	
		13.0	25	*C. asiatica*	very active	[67]
		23.0	50		very active	
		26.0	75		very active	
		28.0	100		very active	
	*Pseudomonas aeruginosa*	8.0	64	*S. robusta*	active	[62]
		6.0	75	*C. asiatica*	inactive	[67]
		6.0	100			

* purchased from Sigma-Aldrich Chemicals Inc.; ^C^—origin is not given.

Table 2 presents the results of the research in which the authors marked the zones of inhibition of bacterial growth in the presence of selected AA concentrations. The largest zones of growth inhibition were recorded for *K. pneumoniae* (13–28 mm) using AA in concentrations of 25–100 mg/L [67]. The higher the concentration, the greater the zone of inhibition of bacterial growth. Very good AA activity was also demonstrated against *B. subtilis* using concentrations of 75 and 100 mg/L and *S. aureus* by treating it with AA at a concentration of 100 mg/L [67]. In the case of *B. cereus*, only a very high concentration of AA (2000 mg/L) isolated from *M. malabathricum* leaves showed antibacterial activity [66]. On the other hand, the smallest zone of growth inhibition was observed for the *P. aeruginosa* strain exposed to AA at the concentrations of 75 and 100 mg/L [67]. The diameter of this zone was only 6 mm, which proved the lack of antibacterial activity of the acid. In most of the described cases, AA showed zone inhibition was 7–12 mm against Gram-positive bacteria: *B. cereus* [66], *B. subtilis* [67], *S. aureus* [66,67,68], and *S. pneumoniae* [68] and Gram-negative: *E. coli* [62,68], *H. pylori* [68], *K. pneumoniae* [66], and *P. aeruginosa* [62].

Norzaharaini et al. [68] investigated the antibacterial activity of AA and asiaticoside present in the leaves of *C. asiatica*. At concentrations of 500–1000 mg/L, AA had antimicrobial activity against *H. pylori* ATCC 45903, *E. coli* ATCC 29952, *S. pneumoniae*, and *S. aureus*. The disk diffusion method was used for the assessment of antibacterial activity; the zones of inhibition were 12 mm (1000 mg/L) and 8 mm (500 mg/L) for *H. pylori*, 8 mm (1000 mg/L) for *S. aureus*, and 7 mm (1000 mg/L) for *E. coli* and *S. pneumoniae*. On the other hand, asiaticoside did not show antimicrobial activity against any of the above-mentioned strains.

Wong et al. [66], using the agar diffusion method, proved that AA derived from methanolic extract of *M. malabathricum* leaves exhibited antibacterial activity against *B. cereus* ATCC 10876, *S. aureus* ATCC 25923, and *K. pneumoniae*. For AA at a concentration of 2000 mg/L, the zones of inhibition of the analysed strains were 13.5, 11.0, and 9.0 mm, respectively, while for AA at a concentration of 1000 mg/L they were 12.0, 10.5, and 8.0 mm, respectively. AA at concentrations of 500 and 250 mg/L inhibited the growth of *B. cereus* (zone diameter was 11.5 and 10.0 mm, respectively) and *S. aureus* (10.0 and 9.0 mm), however, it was not active against *K. pneumoniae*. None of the used AA concentrations had antibacterial activity against *B. subtilis*, *E. coli* ATCC 25922, *P. aeruginosa* ATCC 17853, and *S. typhi*.

Ashella and Fleming [67] conducted research concerning the antibacterial activity of AA using the agar diffusion-well method against *K. pneumoniae*, *P. aeruginosa*, *B. subtilis*, and *S. aureus* by identifying zones of bacterial growth inhibition at different acid concentrations (25, 50, 75, and 100 mg/L). *K. pneumoniae* proved to be the most susceptible to AA, as the zone of inhibition of growth of these bacteria appeared already at the lowest AA concentration (25 mg/L) and it was 13 mm. The *P. aeruginosa* strain was the most resistant to AA, as a zone of growth inhibition of 6 mm appeared only after using the concentration of 100 mg/L.

In the study by Bharitkar et al. [62], AA isolated from *S. robusta* resin showed antimicrobial activity against *E. coli* ATCC 25938 and *P. aeruginosa*. The zones of inhibition for AA at the same concentration of 64 mg/L were 7 and 8 mm, respectively.

### 6.2. Anti-Planktonic Activity of Ursolic Acid

Many articles describe the activity of UA against planktonic forms of bacteria. UA used in these experiments was plant-derived as shown in Figure 4 or was synthetic.

These articles include a very diverse range of research, for example, determination of MIC values (Table 3), the size of the growth inhibition zones (Table 4), effects of this acid on the cell morphology and virulence traits, the expression of genes associated with various metabolic pathways. There are also articles concerning the synergism of action with UA and antibiotics.

The data presented in Table 3 show that UA shows differentiated antibacterial activity. It is worth noting that this acid showed very good activity only against Gram-positive bacteria. UA isolated from *A. scholaris* showed significant activity against *E. faecalis* (MIC = 1 mg/L) and *L. monocytogenes* (MIC = 2 mg/L) [69]. UA isolated from *S. officinalis* also showed very good activity (MIC = 4 mg/L) against *E. faecalis* and *E. faecium* [70]. On the other hand, synthetic UA showed very good activity (2 ≤ MIC ≤ 7.8 mg/L) against cocci: *S. mutans*, *S. sobrinus* and *S. gordonii* [14,71], *E. faecalis* (MIC = 4 mg/L) [72], *S. aureus* (4 ≤ MIC ≤ 8 mg/L) [72,73,74] and *S. epidermidis* (MIC = 4 mg/L) [75], as well as against *L. monocytogenes* (6.5 ≤ MIC ≤ 8 mg/L) [75,76,77]. The MIC value of 8 mg/L for AU isolated from *A. scholaris* [69], *S. officinalis* [70] and *V. paradoxa* [78] was determined for *B. cereus* [69], *S. aureus* [70,78] and *S. pneumoniae* [70]. UA showed moderate activity (10 < MIC < 100 mg/L) against 13 species of Gram-positive bacteria [46,54,63,69,70,72,73,75,78,79,80,81,82,83,84,85,86,87] and 6 species of Gram-negative bacteria [75,79,82]. On the other hand, the low activity of UA (MIC > 100 mg/L) was described for 10 species of Gram-positive bacteria [13,46,63,72,80,88,89,90,91] and 8 species of Gram-negative bacteria [10,47,63,69,70,72,82,84,88,92,93]. The lowest sensitivity to UA (MIC ≥ 1000 mg/L) showed 5 species of Gram-positive bacteria: *B. cereus* [82], *L. monocytogenes* [82], *S. aureus* [82], *S. mutans* [94], *S. pneumoniae* [80], and 5 species of Gram-negative bacteria: *A. caveae* [82], *E. coli* [10,47,82,95], *K. pneumoniae* [80], *S. choleraesuis* [80], *V. cholerae* [80,82].

**Table 3 biomolecules-12-00098-t003:** MIC values of ursolic acid against Gram-positive and Gram-negative bacteria.

Bacterial Group	Species	Ursolic Acid			Ref.
		MIC [mg/L]	Source	Antibacterial Activity	
Gram-positive	*Actinomyces naeslundii*	16	purchased *^A^	good	[89]
	*Actinomyces viscosus*	32	purchased *^A^	good	[89]
	*Bacillus cereus*	8	*A. scholaris*	very good	[69]
		20	*M. ligustroides*	good	[80]
		≥1024	*S. australis*	weak	[82]
	*Bacillus sphaericus*	50	*D. melanoxylon*	good	[79]
	*Bacillus subtilis*	25	*D. melanoxylon*	good	[79]
	*Enterococcus faecalis*	1	*A. scholaris*	very good	[69]
		4	*S. officinalis*	very good	[70]
		4–256	purchased *^A^	very good–weak	[72]
		16	*S. lancifolia*	good	[63]
		32–512	purchased *^A^	good–weak	[46]
		50	*M. fallax*	good	[87]
		250	*M. ligustroides*	weak	[80]
	*Enterococcus faecium*	4	*S. officinalis*	very good	[70]
		256	purchased *^A^	weak	[72]
		500	*C. macrocarpa*	weak	[88]
	*Listeria monocytogenes*	2	*A. scholaris*	very good	[69]
		6.5	purchased *^A^	very good	[76]
		8	purchased *^A^	very good	[75]
		8	purchased *^A^	very good	[77]
		≥1024	*S. australis*	weak	[82]
	*Staphylococcus aureus*	4–8	purchased *^A^	very good	[74]
		7.8 and 15.6	purchased *^A^	very good, good	[73]
		8	*S. officinalis*	very good	[70]
		8–16	*V. paradoxa*	very good–good	[78]
		8–>256	purchased *^A^	very good–weak	[72]
		10	*B. dracunculifolia*	good	[81]
		10	purchased *^A^	good	[75]
		16	*A. scholaris*	good	[69]
		32 and ≥1024	*S. australis*	good, weak	[82]
		37	natural products	good	[85]
		50	*D. melanoxylon*	good	[79]
		60	natural products	good	[86]
		64	*A. scholaris*	good	[69]
		64	purchased *^A^	good	[83]
		64	purchased **^A^	good	[84]
		64	purchased *^A^	good	[54]
		128	*S. lancifolia*	weak	[63]
		250	*C. macrocarpa*	weak	[88]
		250	purchased *^A^	weak	[90]
	*Staphylococcus epidermidis*	7.5	purchased *^A^	very good	[75]
	*Staphylococcus saprophyticus*	250	*C. macrocarpa*	weak	[88]
	*Streptococcus mitis*	50	*M. fallax*	good	[87]
	*Streptococcus gordonii*	7.8	purchased ***^A^	very good	[14]
		64	purchased *^A^	good	[89]
	*Streptococcus mutans*	2–4	purchased *^A^	very good	[71]
		7.8	purchased ***^A^	very good	[14]
		80	*M. fallax*	good	[87]
		128–256	purchased *^A^	weak	[91]
		250	purchased ****^A^	weak	[13]
		256	purchased *^A^	weak	[89]
		1024	purchased *^A^	weak	[94]
	*Streptococcus pneumoniae*	8	*S. officinalis*	very good	[70]
		1000	*M. ligustroides*	weak	[80]
	*Streptococcus salivarius*	50	*M. fallax*	good	[87]
	*Streptococcus sanguinis*	7.8	purchased ***^A^	very good	[14]
		50	*M. fallax*	good	[87]
		128	purchased *^A^	weak	[89]
	*Streptococcus sobrinus*	2–4	purchased *^A^	very good	[71]
		50	*M. fallax*	good	[87]
		64	purchased *^A^	good	[89]
		128	purchased *^A^	weak	[91]
Gram-negative	*Aeromonas caveae*	≥1024	*S. australis*	weak	[82]
	*Escherichia coli*	50	*D. melanoxylon*	good	[79]
		64 and 512	*S. australis*	good, weak	[82]
		>128	*S. officinalis*	weak	[70]
		>128	*S. lancifolia*	weak	[63]
		>128	*A. scholaris*	weak	[69]
		250	*C. macrocarpa*	weak	[88]
		256	purchased **^A^	weak	[84]
		256	purchased *^A^	weak	[93]
		>256	purchased *^A^	weak	[72]
		>256	*V. macrocarpon*	weak	[92]
		512–>1024	purchased *^A^	weak	[10]
		512–>1024	purchased *^A^	weak	[47]
		1000	*E. tereticornis*	weak	[95]
		(1024	*S. australis*	weak	[82]
	*Klebsiella pneumoniae*	64	*S. australis*	good	[82]
		500	*C. macrocarpa*	weak	[88]
		512	purchased *^A^	weak	[93]
		1000	*M. ligustroides*	weak	[80]
	*Pseudomonas aeruginosa*	22.5	purchased *^A^	good	[75]
		>128	*S. officinalis*	weak	[70]
		>128	*S. lancifolia*	weak	[63]
		>128	*A. scholaris*	weak	[69]
		256	purchased **^A^	weak	[84]
		≥256	purchased *^A^	weak	[72]
		500	*C. macrocarpa*	weak	[88]
		512	*S. australis*	weak	[82]
	*Pseudomonas syringae*	25	*D. melanoxylon*	good	[79]
	*Salmonella choleraesuis*	1000	*M. ligustroides*	weak	[80]
	*Salmonella enterica*	>128	*A. scholaris*	weak	[69]
	*Salmonella typhi*	50	*D. melanoxylon*	good	[79]
	*Serratia marcescens*	>128	*S. officinalis*	weak	[70]
	*Shigella flexneri*	64	*S. australis*	good	[82]
	*Vibrio cholerae*	>1000	*M. ligustroides*	weak	[80]
		≥1024	*S. australis*	weak	[82]

* purchased from Sigma-Aldrich Chemicals Inc.; ** purchased from Santa Cruz Biotechnology; *** purchased from Macklin Inc.; **** purchased from National Institutes for Food and Drug Control; ^A^—synthetic.

Table 4 presents the results of the research in which the authors marked the zones of bacterial growth inhibition in the presence of selected concentrations of UA. The largest zones of growth inhibition were recorded for Gram-positive *S. aureus* (21 mm) and Gram-negative *S. boydii* (19 mm) treated with UA (100 mg/L) isolated from *H. corymbosa*, which proves its high activity antibacterial action against these bacteria [96]. On the other hand, UA isolated from *M. malabathricum* [66] and used in concentrations of 250–2000 mg/L showed good activity against Gram-positive bacteria: *B. subtilis* (8.0–10.5 mm) and *S. aureus* (7.5–10.5 mm) and Gram-negative *S. typhi* (9.5–11.0 mm). The higher the acid concentration, the larger the zone of inhibition of growth. Interestingly, this relationship was not observed in the case of *B. cereus*, where the zone of growth inhibition was 7.0 mm regardless of the concentration of UA (250–2000 mg/L) [66].

**Table 4 biomolecules-12-00098-t004:** Growth inhibition zones and corresponding concentration values of ursolic acid against Gram-positive and Gram-negative bacteria.

Bacterial Group	Species	Inhibition Zone [mm]	Ursolic acid			Ref
			Concentration [mg/L]	Source	Antibacterial Activity	
Gram-positive	*Bacillus cereus*	7.0	250	*M. malabathricum*	active	[66]
		7.0	500		active	
		7.0	1000		active	
		7.0	2000		active	
	*Bacillus subtilis*	8.0	250	*M. malabathricum*	active	[66]
		9.0	500		active	
		9.5	1000		active	
		10.5	2000		active	
	*Staphylococcus aureus*	21.0	100	*H. corymbosa*	very active	[96]
		7.5	250	*M. malabathricum*	active	[66]
		8.5	500		active	
		9.5	1000		active	
		10.5	2000		active	
Gram-negative	*Pseudomonas aeruginosa*	10.0	100	*H. corymbosa*	active	[96]
	*Salmonella typhi*	9.5	250	*M. malabathricum*	active	[66]
		10.0	500		active	
		10.0	1000		active	
		11.0	2000		active	
	*Shigella boydii*	19.0	100	*H. corymbosa*	very active	[96]

When analysing the data contained in Table 1 and Table 3, one can observe that both AA and UA showed better antibacterial activity against Gram-positive strains. This is probably related to the differences in the structure of cellular envelopes. The structure present in Gram-negative bacteria is the outer membrane, with a specific chemical structure that makes it difficult for the penetration of antimicrobial compounds into the cell. The analysis of the data in Table 1 and Table 3 also shows that the acid MIC values for the individual strains were generally higher in the case of UA than AA. The more efficient action of AA is probably due to the presence of an additional hydroxyl group at C-23, which makes the whole molecule more hydrophilic and makes it easier to reach the interior of the bacterial cells. It is also worth noting that bacteria belonging to the same group (Gram-positive/Gram-negative), and even to the same species, can significantly differ in their susceptibility to the same antibacterial compounds, resulting in different MIC values. The reasons for these differences may also be a different source/origin of microorganisms (clinical, environmental, or reference strains) and their individual characteristics. Moreover, the antibacterial activity of the acid may be related to the method of obtaining it (natural or synthetic). When analysing the data contained in Table 2 and Table 4, it cannot be unequivocally shown that AA and UA exhibited better antibacterial activity against Gram-positive bacteria than against Gram-negative bacteria.

Mallavadhani et al. [79] investigated the effects of UA and its synthetic lipophilic derivatives containing ester chains of 3-*O*-fatty acids (C12–C18) against various bacterial species. UA was isolated from the leaves of *D. melanoxylon*. The MIC values of UA were 25 mg/L against *B. subtilis* ATCC 6051 and *P. syringae* ATCC 13457, while 50 mg/L against *B. sphaericus* ATCC 14577, *S. aureus* ATCC 9144, *E. coli* ATCC 25922, and *S. typhi* ATCC 23564. It should be noted that the antibacterial activity of UA derivatives against the analysed strains varied and it was dependent on the bacterial species, however, only in a few cases, it was better than that of pure UA.

Scalon Cunha et al. [87] evaluated the antibacterial activity of UA isolated from *M. fallax* and its semi-synthetic derivatives compared to the strains involved in plaque and caries formation: *E. faecalis* ATCC 4082, *S. salivarius* ATCC 25975, *S. mitis* ATCC 49456, *S. mutans* ATCC 25275, *S. sobrinus* ATCC 33478 and *S. sanguinis* ATCC 10556. The MIC values of UA against all the aforementioned strains were in the range of 50–80 mg/L. In contrast, the MIC values of UA derivatives were generally higher. The stronger antibacterial activity of UA compared to its derivatives suggests that the free hydroxyl group at C-3 and the carboxyl group at C-17 are primarily responsible for this activity.

According to the study by Horiuchi et al. [70], UA exhibited antibacterial activity against Gram-positive cocci such as methicillin-resistant *S. aureus* (MRSA), penicillin-resistant *S. pneumoniae* (PRSP), and vancomycin-resistant enterococci (VRE): *E. faecalis* and *E. faecium*. The MIC values were 8, 8, and 4 mg/L, respectively. At a concentration of 2 × MIC, UA showed the bactericidal activity against VRE for at least 48 h. UA showed a stronger bactericidal activity against *E. faecium* than ampicillin. UA showed only a bacteriostatic activity against *E. faecalis*. However, UA was not active against Gram-negative bacteria: *P. aeruginosa*, *E. coli*, *S. marcescens*, or against mutants of *E. coli* and *P. aeruginosa*, which are hypersensitive to drugs due to the absence of multidrug efflux pumps (MDR pumps). The MIC value against those strains was >128 mg/L.

The study by Ahmad et al. [97] determined the antimicrobial activity of UA isolated from the aerial parts of *M. elegans* against *B. subtilis*, *S. aureus*, *P. aeruginosa*, *S. typhi*, *E. coli*, and *S. flexneri*. UA revealed the antibacterial activity against *B. subtilis*, *S. aureus*, *P. aeruginosa*, *S. typhi* for which growth inhibition zones were 12, 15, 16, and 18 mm, respectively. UA did not inhibit the growth of the other two strains. 

Fontanay et al. [72] determined MIC for UA against five reference strains: *E. coli* ATCC 25922, *S. aureus* ATCC 25923 and ATCC 29213*, E. faecalis* ATCC 29212 and *P. aeruginosa* ATCC 27853, as well as five antibiotic-resistant clinical isolates: *E. coli*, *S. aureus*, *E. faecium*, *E. faecalis*, and *P. aeruginosa*. UA was found to be highly effective against Gram-positive bacteria: *E. faecalis* ATCC 29212 (MIC = 4 mg/L) and *S. aureus* ATCC 25923, *S. aureus* ATCC 29213 (MIC = 8 mg/L). UA had no antibacterial activity against *E. coli* ATCC 25922, *P. aeruginosa* ATCC 27853, and all clinical strains (MIC > 256 mg/L).

Da Silva Filho et al. [81] showed that UA isolated from *B. dracunculifolia* leaves had antimicrobial activity against *S. aureus* ATCC 43300 (MRSA). The growth inhibitory concentration for 50% was IC50 = 5 mg/L and the MIC value was 10 mg/L. 

Huang et al. [92] identified the presence of UA in one fraction of an extract prepared from *V. macrocarpon* cranberry fruit. The MIC value of the extract determined against strains of uropathogenic *E. coli* ATCC 700336 and ATCC 25922 was >256 mg/L. The extract fraction at 10 mg/L also inhibited COX-2 activity and the activity of nuclear transcription factor, NF-κβ.

Cunha et al. [80] evaluated the activity of UA isolated from *M. ligustroides* and its derivatives against *B. cereus* ATCC 14579, *V. cholerae* ATCC 9458, *S. choleraesuis* ATCC 10708, *K. pneumoniae* ATCC 10031, and *S. pneumoniae* ATCC 6305. UA was active only against *B. cereus*, showing the MIC value of 20 mg/L; the MIC values for other strains were 1000 mg/L or higher. UA derivatives were effective only against *S. pneumoniae*; their MIC values were 50 mg/L.

According to Kurek et al. [76], UA improved the lytic activity of Triton X-100 and lysozyme against the strain of *L. monocytogenes*. After 24-h incubation in the presence of UA at a concentration of 0.7 × MIC (4.5 mg/L), bacterial cell length was reduced by 20% compared to the control and it did not exceed 2 μm.

Sultana et al. [96] studied the antibacterial activity of UA isolated from *H. corymbosa* against 4 strains of Gram-positive bacteria: *B. subtilis*, *B. cereus*, *S. lutea*, and *S. aureus*, as well as 10 Gram-negative strains: *S. sonnei*, *S. dysenteriae*, *S. boydii*, *S. paratyphi A*, *E. coli*, *P. aeruginosa*, *S. typhi*, *S. flexneri*, *V. cholereae*, and *K. pneumonia*. UA at 100 mg/L showed significant activity against *S. aureus* (21 mm diameter zone of inhibition) and *S. boydii* (19 mm), whereas a moderate activity against *P. aeruginosa* (10 mm). Unfortunately, a weak activity of UA against other microorganisms (zones of inhibition of 6–10 mm) was reported.

Acebey-Castellon et al. [63] found that UA derived from *S. lancifolia* leaves showed a stronger antibacterial activity against Gram-positive *S. aureus* ATCC 25923 and *E. faecalis* ATCC 29212, and a weaker activity against Gram-negative *E. coli* ATCC 25922 and *P. aeruginosa* ATCC 27853. The MIC values ranged from 16 to >128 mg/L and they were lowest against *E. faecalis*. 

Filocamo et al. [73] analysed the antibacterial activity of UA (also combined with norfloxacin) against *S. aureus* ATCC 29213, *S. aureus* ATCC 43300 (MRSA), and 42 clinical isolates of *S. aureus*. The MIC and MBC values of UA against *S. aureus* ATCC 29213 were 7.81 mg/L and 124.96 mg/L, respectively. MRSA strain was more resistant to UA, the MIC and MBC values were 15.62 mg/L and 249.92 mg/L, respectively. The synergistic effect of UA and the antibiotic was proved by the combination of 0.25 × MIC of norfloxacin with 1 × MIC of UA, which inhibited the growth of 21 of 42 clinical strains, while the combination of 0.5 × MIC of norfloxacin with 2 × MIC of UA inhibited as many as 90% of the strains. Against *S. aureus* strain ATCC 29213, UA at a concentration of 1 × MIC showed no postantibiotic effect (PAE), while at a concentration of 2 × MIC it exerted a very short PAE (2 h). Administration of UA combined with norfloxacin resulted in the prolongation of PAE up to 14.5 h (at 1 × MIC of UA) and 21 h (at 2 × MIC of UA). There was a significant prolongation (25.3–29.1 h) of the postantibiotic sub-MIC effect (PASME) through preincubation of *S. aureus* ATCC 29213 cells in sub-inhibitory concentrations of norfloxacin. The synergistic effect of UA and norfloxacin may be due to initial damage to the bacterial membrane caused by lipophilic UA, which increases its permeability to norfloxacin. Enhanced intracellular accumulation of norfloxacin results in increased bacterial DNA damage and thus longer PAE and PASME.

Kim et al. [71] determined the antibacterial activity of UA against *S. mutans* ATCC 25175, *S. sobrinus* ATCC 33478, 40 clinical strains of *S. mutans*, and 15 clinical strains of *S. sobrinus*, which are involved in dental caries. The MIC value of UA against both reference strains was 2 mg/L while the MBC values were >4 mg/L for *S. mutans* ATCC 25175 and 8 mg/L for *S. sobrinus* ATCC 33478. The MIC value of UA was 2 mg/L for most clinical strains. The exceptions were 1 strain of *S. mutans* and three strains of *S. sobrinus* (MIC = 4 mg/L). Based on the obtained results, the authors suggested that UA at concentrations >8 mg/L could be used as an ingredient in oral hygiene products to prevent dental caries.

Moodley et al. [88] used UA isolated from the leaves of *Carissa macrocarpa* in their study. The acid revealed a moderate antibacterial activity against three Gram-negative bacterial strains: *K. pneumoniae* ATCC 700603, *P. aeruginosa* ATCC 35032, *E. coli* ATCC 25922, as well as four Gram-positive strains: *S. saprophyticus* ATCC 35552, *S. aureus* ATCC 25923, *S. aureus* ATCC 43300, and *E. faecium* ATCC 19434. The MIC values of that acid were 250–500 mg/L. Moodley et al., in their study, also identified the effect of UA at concentrations of 0.5 × MIC, 1 × MIC, and 2 × MIC on the adhesion of all the above-listed strains to polystyrene surfaces. The adhesion of *E. coli* and *S. aureus* cells was reduced under the influence of all acid concentrations used. In the case of *P. aeruginosa*, there was reduced adhesion only with exposures to UA at 0.5 × MIC and 1 × MIC concentrations while there was enhanced adhesion during the exposure to 2 × MIC. On the other hand, the adhesion of *S. saprophyticus* was enhanced during exposure to 0.5 × MIC and it was reduced under the influence of 1 × MIC and 2 × MIC. It should be noted that the adhesion of *K. pneumoniae* to polystyrene surfaces enhanced after their exposure to all UA concentrations under analysis.

Kim et al. [74] identified the antibacterial activity of UA against 19 clinical strains of MRSA. The MIC value of UA for the analysed 15 strains was 4 mg/L while for the remaining strains—8 mg/L. Greater variation was observed for MBC values, which ranged between 4–32 mg/L.

According to the study by Kurek et al. [75], UA revealed significant antimicrobial activity against planktonic cells of strains such as *P. aeruginosa, L. monocytogenes, S. aureus* ATCC 29213, and *S. epidermidis* ATCC 12228. MIC values ranged from 7.5 mg/L for *S. epidermidis* to 22.5 mg/L for *P. aeruginosa*. Furthermore, the authors demonstrated the synergism of action of UA combined with β-lactam antibiotics (ampicillin and oxacillin). It should be noted that bacteria that were cultured in the presence of both components (UA + oxacillin or UA + ampicillin) became more susceptible to each of them. The most significant effect was observed for *S. aureus* and UA + ampicillin combination. The MIC value for ampicillin was reduced as much as 16-fold, that is, from 4 mg/L to 0.25 mg/L. These results indicate that UA when administered in combination with β-lactams, may be useful in the treatment of bacterial infections.

Wong et al. [66] proved using the agar diffusion method that UA derived from the methanolic extract of *M. malabathricum* leaves exhibited antimicrobial activity against *S. aureus* ATCC 25923, *B. subtilis, B. cereus* ATCC 10876, and *S. typhi*. For UA at 2000 mg/L, the zones of inhibition of the tested strains were 10.5, 10.5, 7.0, and 11.0 mm; for UA at 1000 mg/L, they were 9.5, 9.5, 7.0, and 10.0 mm; for UA at 500 mg/L, they were 8.5, 9.0, 7.0, and 10.0 mm; and for UA at 250 mg/L, they were 7.5, 8.0, 7.0 and 9.5 mm, respectively. UA was not active against *E. coli* ATCC 25922, *P. aeruginosa* ATCC 17853, and *K. pneumoniae* at any of the analysed concentrations.

Wojnicz et al. [47], in addition to the previously described antimicrobial properties of AA, also investigated the effects of UA on the survival and virulence factors of 20 clinical UPEC strains isolated from the urine of patients with pyelonephritis. The MIC value of UA against 18 of 20 analysed bacilli was ≥1024 mg/L. The MIC of that acid was 512 mg/L for only two strains. It was also found that the reduction in the growth of planktonic forms was dependent on both incubation time and UA concentration. UA at a concentration of 40 mg/L significantly reduced the survival of planktonic forms only after 24 h, while at a concentration of 50 mg/L as early as in a 6-h culture. Moreover, UA revealed significant effects on virulence traits of bacilli such as P fimbriae, curli fimbriae, and the ability to synthesise α-hemolysin. The lowest used UA concentration (10 mg/L) resulted in a loss of haemagglutinating capacity associated with the presence of P fimbriae in 20% of the bacilli under analysis. The loss of curli fimbriae, the anti-hemolytic effect, and impaired bacterial motility were observed only after using UA at concentrations of 40 µg/mL and 50 mg/L. The other study [10] reports the effect of UA on surface hydrophobicity and adhesion of clinical UPEC strains to uroepithelial cells. According to the authors, UA reduced the hydrophobic nature of the bacterial cell surface. At the highest concentration used (50 mg/L), UA significantly reduced the number of *E. coli* bacilli adhering to uroepithelial cells. That acid, at a concentration of 50 mg/L, also induced morphological changes in bacterial cells. In UA-treated cultures, there were increased numbers of short filaments (5–15 μm). There were also long filaments (>15 μm), thickened cells with swollen filaments, and cells partially devoid of a cell wall (“ghost cells”).

Do Nascimento et al. [82] determined MIC values for UA and its derivatives isolated from *S. australis* against 12 bacterial strains. UA revealed the strongest activity against *S. aureus* ATCC 6538 (MIC = 32 mg/L). UA was also effective against *E. coli* ATCC 25922, *K. pneumoniae* ATCC 10031, and *S. flexneri* ATCC 12022 (MIC = 64 mg/L). In the case of *E. coli* ATCC 27 and *P. aeruginosa* ATCC 15442, the MIC value was identical (512 mg/L). The six other strains, that is, *S. aureus* ATCC 12692, *S. aureus* ATCC 12624, *B. cereus* ATCC 33018, *A. caveae* ATCC 15468, *V. cholera* ATCC 15748, and *L. monocytogenes* ATCC 19117 were insusceptible to UA (MIC ≥1024 mg/L). The researchers also found the synergism of action of UA combined with aminoglycosides (neomycin, amikacin, kanamycin, and gentamicin). MIC values of those antibiotics were reduced in the presence of UA against almost all microorganisms, excluding gentamicin and amikacin against *E. coli* ATCC 25922 and kanamycin against *P. aeruginosa* ATCC 15442 and *K. pneumoniae* ATCC 10031.

Dwivedi et al. [95] investigated the antibacterial properties of UA isolated from *E. tereticornis* leaves and its lipophilic derivatives. Three strains were used in a study by Dwivedi et al.: nalidixic acid-sensitive *E. coli* CA8000, nalidixic acid-resistant *E. coli* DH5α, and the MDREC-KG4 clinical isolate of *E. coli* that is resistant to multiple antibiotics, including tetracycline. The MIC value of UA against all *E. coli* strains was 1000 mg/L. Similarly, the MIC values of lipophilic derivatives against bacilli were very high (500–1000 mg/L), indicating their lack of antibacterial activity. However, it should be emphasised that the authors found the synergistic effect of both UA and its derivatives combined with nalidixic acid against *E. coli* CA8000 and *E. coli* DH5α, as well as with tetracycline against *E. coli* MDREC-KG4. When applied at a concentration of 10 mg/L, UA and its derivatives combined with nalidixic acid reduced the MIC values of this antibiotic against *E. coli* CA8000 and *E. coli* DH5α by 2-fold (UA) and 4–8-fold (UA derivatives). On the other hand, UA and its derivatives at a concentration of 50 mg/L and combined with tetracycline, reduced the MIC values of this antibiotic against *E. coli* MDREC-KG4 strain by 2-fold and 8-fold, respectively.

Park et al. [98] determined the effect of UA (64 mg/L) on the expression of 22 genes that are important in all three steps of peptidoglycan biosynthesis in *S. mutans* UA159. UA inhibited transcription of all genes involved in the first step (*glmU, murA, murB, murC, murC2, murD, murE, arl, ddl, murI, murF*) and second step of peptidoglycan biosynthesis (*bacA*, *mraY*, *murG*, *murM*, *murN*), and of most genes involved in the third step of murein synthesis (*pbp2a*, *pbp2b*, *pbp2x*, *dacA*). These studies clearly indicate that inhibition of gene expression is one of the mechanisms of the antimicrobial activity of UA.

A study by Wang et al. [69] revealed good antibacterial activity of UA, isolated from *A. scholaris* leaves, against Gram-positive strains of *E. faecalis* ATCC 29212 (MIC = 1 mg/L), *L. monocytogenes* ATCC 7644 (MIC = 2 mg/L), *B. cereus* ATCC 9139 (MIC = 8 mg/L), methicillin-susceptible *S. aureus* (MSSA) ATCC 29213 (MIC = 16 mg/L) and *S. aureus* ATCC 43300 (MRSA) (MIC = 64 mg/L), while a weaker antibacterial activity (MIC >128 mg/L) was against Gram-negative *E. coli* ATCC 35150, *P. aeruginosa* ATCC 27853 and *S. enterica* ATCC 13311. Synergism was also found in the joint action of 0.5 × MIC of UA and 0.5 × MIC of ampicillin or tetracycline against *B. cereus* strains, MSSA and MRSA. The reduction in viable cell counts in 24-h cultures was >4 log10. 

Wang et al. [83] investigated the effect of supra-inhibitory UA concentration on cell membrane integrity and changes in the expression of 29 proteins involved in transcription, translation, and various metabolic pathways in MRSA strain cells. The MIC value of UA was 64 mg/L. After the cocci had been exposed to UA at a concentration of 4 × MIC, the bacteria retained 49.5% of their cell membrane integrity. UA enhanced the synthesis of 18 proteins, of which the highest expression was observed for RplU (translation), AhpC (oxidative stress protein), ClpC (protein folding and RNA degradation processes), Mqo2 (tricarboxylic acid cycle), and Adh (alcohol dehydrogenase). The obtained results indicate that UA, due to its pleiotropic activity against MRSA, is a promising antimicrobial agent that should be given more attention in pharmaceutical research.

UA isolated from *V. paradoxa* leaves revealed good antimicrobial activity against reference *S. aureus* ATCC 33591 and clinical MRSA strains [78]. The MIC values of that acid were 8–16 mg/L and they were significantly lower than the MIC values of ampicillin and oxacillin (32–512 mg/L). The authors also found the presence of synergism between UA and β-lactam antibiotics (ampicillin and oxacillin), resulting in reduced MIC values for both drugs. The obtained results suggest a bidirectional mechanism of UA action. As it is known, a penicillin-binding protein (PBP2A) and β-lactamases are involved in the resistance of methicillin in staphylococci. Therefore, the authors suggested that UA might restore the affinity of PBP2A protein to β-lactam antibiotics and restore β-lactamase activity. Through observation by a fluorescence microscope, the authors proved that UA, like oxacillin, induces the delocalisation of PBP2 from the site of a dividing septum and its redistribution within the cell membrane, leading to disruption of cell division.

Oloyede et al. [84] found that the antimicrobial activity of UA is directly due to its oxidative properties. Preliminary studies identified the antimicrobial activity of UA against *E. coli* ATCC 25922, *P. aeruginosa* ATCC 27853, and *S. aureus* ATCC 29213 strains. The MIC values against these bacteria ranged from 64–256 mg/L, while the MBC values ranged from 256–512 mg/L. According to further studies, UA can generate the production of reactive oxygen species (ROS), especially superoxide anion radical and hydroxyl radical, in bacterial cells. They cause enhanced electron transport chain activity, resulting in oxidative stress, lipid peroxidation, and oxidation of 2-deoxyribose in DNA, leading to bacterial cell death. Increased amounts of ROS are associated with dysfunctions in the conversion reactions of glutathione, a natural free radical scavenger. The authors proved that UA decreased the level of reduced glutathione in bacterial cells while increasing the number of its oxidised molecules, which contributed to a significant decrease in bacterial survival due to an increase in oxidative stress indices.

Wojnicz et al. [46], in addition to their previously presented research concerning the effect of AA on clinical uropathogenic *E. faecalis* strains, also analysed the antibacterial effect of UA against cocci that are a common cause of UTIs, especially in catheterised patients. The bacteria were resistant to gentamicin, nitrofurantoin, ampicillin, and trimethoprim/sulphamethoxazole. The MIC values of UA against *E. faecalis* strains were 32–512 mg/L, which corresponds to other available literature data. It was noted that the growth of planktonic forms of *E. faecalis* was inhibited under the influence of 0.75 × MIC of UA as early as in the 2-h culture, while in the 6-h culture, the greatest 780-fold reduction was observed in the number of CFU/mL of viable cells compared to the control sample. It was also found that in the presence of UA, *E. faecalis* cells were larger and they formed aggregates instead of characteristic chains. It is likely that UA impairs cell division processes, which may cause phenotypic changes in cell morphology, such as increased cell diameter and the presence of irregular aggregates. Moreover, it was found that UA significantly reduced, and in some cases completely inhibited, the ability of *E. faecalis* strains to synthesise DNase and extracellular enzymes (lipase, lecithinase, gelatinase) and haemolysin that damage host tissues.

Zhou et al. [99] investigated the antibacterial activity of ursolic acid 3-*O*-α-*L*-arabinopyranoside (URS), isolated from *A. henryi* leaves, against *S. aureus* strains. The MIC values were found to be 3.125 mg/L against MSSA strain and 6.25 mg/L against MRSA strains. The authors also found the existence of synergism between URS and oxacillin against MRSA strains. The addition of 0.5 × MIC of URS reduced the MIC values of oxacillin by 2–32-fold according to strain. The study also determined the effect of URS on the morphology of MRSA strain cells. URS at a concentration of 0.5 × MIC damaged the bacterial cell membrane and caused surface roughness. The authors also noted a slight decrease in membrane protein PBP2A levels under the influence of URS. A significant decrease in PBP2A levels, cell membrane disintegration, and bacterial cell lysis were only observed in the assay containing the combination of URS with oxacillin.

Jabeen et al. [100] investigated the activity of hydrazide of UA and its 11 derivatives with metals against Gram-negative *S. typhi* and *Shigella* spp., as well as Gram-positive *S. aureus* and *S. pneumoniae*. The MIC value of hydrazide of UA against the above-mentioned strains was >256 mg/L. Furthermore, it was found that the antibacterial activity of the hydrazide of UA might be enhanced by producing complexes with various metals (Zn, Cu, Fe, Sb), then the MIC values were 4–32 mg/L. The complex with triphenyltin proved to be the most potent antimicrobial agent, with the MIC values of 4 mg/L against *S. pneumoniae* and 8 mg/L against other strains.

Sundaramoorthy et al. [93] determined the MIC for UA against extremely drug-resistant (XDR) clinical strains of *E. coli* (MIC = 256 mg/L) and *K. pneumoniae* (MIC = 512 mg/L). The researchers found that UA exhibited the synergism of action with colistin, causing a significant 16-fold reduction in colistin MIC for *E. coli* and a 4-fold for *K. pneumoniae*. Moreover, UA enhanced the permeability of the bacterial outer membrane, and that facilitated the transport of colistin into the bacterial cell. Furthermore, UA inhibited the activity of efflux pumps, which in turn impeded the removal of the antibiotic from the bacterial cell.

When analysing the antibacterial activity of AA or UA, it can be noticed that their antimicrobial activity (MIC/inhibition zone) may be different for the same bacterial species. However, there are several different aspects to consider. First, there are many different bacterial strains within a species, all from different origins (biological material from which it was isolated, e.g., urine, faeces, blood, etc.). Secondly, the acids used in separate studies often have different origins, that is, they are isolated from different plant species. Moreover, the methods of extracting these metabolites often vary.

## 7. Anti-Biofilm Properties of Pentacyclic Triterpenes

It is currently known that more than 99% of both commensal and pathogenic bacteria in the human body occur in a biofilm form. Biofilms are formed on the surface of dead cells (skin microflora), viable cells (mucous membrane microflora), and abiotic surfaces found in the human body (catheters, implants). Consortia formed from pathogenic bacteria are difficult to destroy with antibiotics, so it is useful to learn about the effects of AA and UA on biofilm structure and their interaction with drugs used in the treatment of bacterial infections.

### 7.1. Activity of Asiatic Acid against Bacterial Biofilms

A negligible number of articles have been devoted to the effect of AA on biofilm formation and eradication. Therefore, we decided to analyse available research in the current review in which the antibiofilm properties of AA combined with antibiotics were also highlighted.

Garo et al. [64] investigated the effect of AA and its combinations with ciprofloxacin or tobramycin on single-species *P. aeruginosa* biofilm. A rotating disk reactor (RDR) was used in that study as a model device to study the susceptibility of biofilms to antibiotics and other compounds. The MIC value of AA for *P. aeruginosa* was >128 mg/L. AA at concentrations of 10, 50, and 100 mg/L was used in that study. Only a small degree of biofilm reduction was observed under the influence of AA at concentrations of 50 and 100 mg/L. However, it should be noted that the lowest concentration of AA (10 mg/L) enhanced the susceptibility of biofilm-living *P. aeruginosa* to both tobramycin (100 mg/L) and ciprofloxacin (10 mg/L), which previously showed no antibiofilm effect.

Wojnicz et al. [49] conducted a study determining the effect of AA and its combinations with ciprofloxacin on biofilm formation and eradication. Uropathogenic *E. coli* strains (reference ATCC 700928 strains and clinical strains), with virulence traits relevant to biofilm formation, were used in that study. AA at a concentration of 50 mg/L weakly inhibited the biofilm formation by *E. coli* strains. A significantly better effect was obtained when AA was used in combination with ciprofloxacin. Similarly, during the eradication of biofilm from urological catheters, statistically significant results were obtained only when AA was used in combination with ciprofloxacin. The number of viable bacteria was reduced to 12%. In another study, Wojnicz et al. [46] investigated the effect of 0.75 × MIC of AA on biofilm production by clinical *E. faecalis* strains. The MIC values of AA against the analysed strains were 64–128 mg/L. The complete inhibition of biofilm synthesis by tested cocci was demonstrated throughout the duration of the experiment (1–10 days). The survival rate of bacteria in the biofilm mass was significantly reduced under the influence of AA compared to control samples. 

Surprising results were obtained by Harnvoravongchai et al. [11]. The research revealed that AA even at a concentration of 80 mg/L (8 × MIC) had no inhibitory effect on biofilm formation by the highly virulent reference strain of *C. difficile* R20291.

The recent study of Sycz et al. [9] revealed that AA decreased the survival and the ability to create single- and multi-species biofilms by uropathogenic *E. coli* CFT073, *E. cloacae* ATCC-BAA 2468, *P. aeruginosa* ATCC 25000 strains. AA also changed the morphology of these bacteria.

### 7.2. Activity of Ursolic Acid against Bacterial Biofilms

Much more attention was paid to the antibiofilm properties of UA compared to AA. The following articles describe the effects of UA on biofilm mass formation and the expression of genes encoding virulence factors associated with biofilm synthesis. Few articles highlighted the synergistic effect of UA and antibiotics on biofilm formation.

Ren et al. [101] analysed the antimicrobial activity of UA isolated from *Diospyros dendo* leaves. UA at a concentration of 10 mg/L already showed significant antimicrobial activity against 24-h single-species biofilms formed by *E. coli* (including ATCC 25404), *P. aeruginosa*, and *V. harveyi* strains, reducing the amount of biofilm mass they produced by 72%, 87%, and 57%, respectively. Interestingly, UA at concentrations of 10 and 30 mg/L did not inhibit the growth of the aforementioned strains growing in planktonic forms. It was also found that UA did not affect quorum sensing. In contrast, UA at concentrations of 10 and 30 mg/L induced the expression of genes encoding proteins related to chemotaxis (*cheA*, *motAB*, *tap*, *tsr*), heat shock (*hslSTV*, *htpG*, *mopB*), and membrane transport (*dcuA*, *emrK*, *malE*). It should be noted that overexpression of the *motAB* gene makes cells too motile to remain stable within the biofilm environment, resulting in reduced biofilm formation. Conversely, low cell motility caused by loss of the *motAB* gene promotes bacterial conjugation, which promotes biofilm development. The authors found that UA (at concentrations of 10 and 30 mg/L) inhibited the operon cysDJK regulated by CysB. The CysB protein is a transcriptional regulator of LysR that controls the expression of genes involved in cysteine biosynthesis and sulphur metabolism. The CysB pathway is an interesting potential target pathway for UA and other TPs. According to that study, UA can modulate *cysB* gene expression in *E. coli*. The *cysB* mutant enhanced biofilm formation by 2–10-fold compared to the isogenic cysB strain.

According to Kurek et al. [75], single-species biofilms formed by *P. aeruginosa*, *L. monocytogenes*, *S. aureus* ATCC 29213 and *S. epidermidis* ATCC 12228 strains had approx. 4-fold higher resistance to UA, compared to planktonic cells. Administration of UA along with β-lactam antibiotics (ampicillin and oxacillin) reduced MBIC (Minimum Biofilm Inhibitory Concentration) of both UA and antibiotics against the aforementioned bacterial strains.

Kim et al. [102] evaluated the effect of UA on biofilm formed by *S. mutans* UA159 cocci that play a key role in the pathogenesis of dental caries. UA in concentrations of 0.1, 0.2, and 0.5 % (*w*/*w*) was used for saturating the composite resin disks in that study. The disks were soaked in a culture of *S. mutans*, which was incubated for 24 h and then plated on nutrient agar to measure the number of CFU/mL. It was found that the number of bacteria reduced with an increasing concentration of UA. However, it should be noted that a better antibacterial effect of UA was obtained in cultures where glucose was the carbohydrate source than in those containing sucrose. 

Zhou et al. [89] also investigated the effects of UA on caries-forming bacteria living both in planktonic form and in biofilm consortia. In addition to *S. mutans* UA159 strain, the researchers used three other strains from the genus *Streptococcus* (*S. sanguinis* ATCC 10556, *S. gordonii* ATCC 10558, and *S. sobrinus* ATCC 6715), as well as two strains from the genus *Actinomyces* (*A. viscosus* ATCC 15987, *A. naeslundii* ATCC 12104). The MIC values of UA were significantly lower against *Actinomyces* spp. (16 mg/L and 32 mg/L) than its MIC values against *Streptococcus* spp. (64–256 mg/L). The researchers found that UA at the sub-inhibitory concentration (0.25 × MIC) inhibited biofilm formation by *S. mutans* and *S. gordonii* on titer plates as well as *S. mutans* and *A. viscosus* on the tooth surface. The study also attempted to eradicate mature biofilms formed by *S. mutans* and *A. viscosus* from the tooth surface using supra-inhibitory concentrations of UA. A much better effect was obtained against *A. viscosus*, which is largely related to the less compact structure of the biofilm formed by that strain compared to the biofilm formed by *S. mutans*. The exopolysaccharide of *S. mutans*, due to the presence of extracellular matrix glucans, is much denser and much more compact, which presumably made it much more difficult for UA to penetrate the inside of the biofilm formed by these cocci and for the bacteria to survive in central parts of the biofilm.

Kurek et al. [77] investigated the effects of UA on two major virulence factors of *L. monocytogenes*—haemolytic activity and biofilm synthesis. After the MIC (8 mg/L) and MBIC (24 mg/L) values of UA had been measured, the researchers determined the effect of sub-inhibitory concentrations of UA on the production of listeriolysin O by *L. monocytogenes*, biofilm formation ability, and survival of bacteria living in the biofilm mass. It was reported that UA at a concentration of 0.75 × MIC inhibited the activity of listeriolysin O almost by 3-fold. A concentration of 0.5 × MBIC of UA attenuated biofilm formation by more than 60% and it reduced the survival of *L. monocytogenes* cells in biofilms by 56%.

The aim of a study by Micota et al. [90] was to determine the effect of UA on adhesion and biofilm formation by coagulase-positive *S. aureus* strains that are a common cause of infective endocarditis. Titre plates with wells coated with fibrinogen, fibronectin, and collagen were used in that study. UA at a concentration of 0.75 × MIC (187.5 mg/L) significantly reduced the bacterial adhesion to surfaces coated with matrix proteins: collagen by 73.2%, fibronectin by 58.8%, and fibrinogen by 65.9%. The impairment of adhesion activities of staphylococci under the influence of UA contributed to significant inhibition of biofilm formation by these bacteria on analysed surfaces (70–86%).

Qin et al. [85] proved that UA at a concentration of 30 mg/L inhibited biofilm formation by a clinical MRSA strain by 66.3%. However, UA did not eradicate the mature biofilm formed by those bacteria. The researchers also attempted to determine the mechanism of the antibiofilm activity of UA at the molecular level. They investigated the expression levels of key genes encoding virulence factors such as surface proteins, capsule polysaccharides, and other compounds associated with biofilm formation by *S. aureus*. According to the researchers, the presence of UA resulted in reduced expression of genes encoding adhesins (*isdB*, *srtB*, *ebh*, *sdrC*) and some genes related to metabolism (*arcA*, *arcB2*, *arcD*, *aur*), which are considered important for biofilm survival.

Zou et al. [91] determined the synergistic effect of UA and xylitol on biofilm synthesis by *Streptococcus* bacteria that are the main aetiological agent of dental caries in humans. Reference strains of *S. sobrinus* ATCC 33478 and *S. mutans* UA159 as well as 2 clinical strains of *S. mutans* (KCOM 1207 and KCOM 1128) were used in that study. The MIC values of UA were 128–256 mg/L and the MBC values ranged from 256–512 mg/L, according to the analysed strain. Interestingly, both the MIC and MBC values of xylitol were identical regardless of the analysed strain. The most UA-susceptible strains were found to be *S. mutans* KCOM 1207 and *S. sobrinus* ATCC 33478. The synergism of action of UA and xylitol was investigated by using combinations of these two components at different concentrations. Combinations of 20% xylitol with sub-inhibitory concentrations of UA (16 or 32 mg/L) significantly reduced biofilm formation by analysed streptococci.

Gilabert et al. [103] investigated the antimicrobial activity of UA isolated from the liverwort *Lepidozia chordulifera* against reference strains of *P. aeruginosa* ATCC 27853 and *S. aureus* ATTC 6538P. UA at a concentration of 50 mg/L did not decrease the amount of biofilm mass produced by *P. aeruginosa*, but it resulted in a 33% increase in biofilm production by *S. aureus* compared to the control. Moreover, UA had a stimulating effect on the growth of both *P. aeruginosa* and *S. aureus*, increasing the number of these bacteria by 41% and 12%, respectively. It should be noted that despite the lack of antibiofilm properties, UA reduced the activity of elastase (LasB), produced by *P. aeruginosa*, by 96%. It is believed that this enzyme affects biofilm architecture and functionality [104], while inhibition of LasB activity reduces bacterial adhesion, microcolony formation, and EPS binding in the biofilm [105].

Lou et al. [106] conducted a study concerning the effect of different components, isolated from *Arctium lappa* leaves, on the ability of biofilm synthesis by *P. aeruginosa* ATCC 9027. UA, along with rutin, caffeic acid, coumaric acid, and quercetin, was found to be one of the five best-performing antibiofilm substances present in *A. lappa* leaves. The lowest UA concentration that completely inhibited biofilm formation by *P. aeruginosa* was 500 mg/L.

Tan et al. [86] extended the research by Qin et al. [85] concerning the identification of the antibiofilm mechanism of UA activity at the molecular level. A reference strain of *S. aureus* ATCC 2592 (MSSA), which can form a vancomycin-resistant biofilm, was used in that study. The identified MIC and MBC values for UA were 60 mg/L and >200 mg/L, respectively. It was found that UA inhibited biofilm mass growth by 46.5%. The expression of six genes (*agrA*, *hld*, *icaR*, *spa*, *cna*, *bbp*) involved in biofilm formation in UA-treated bacteria was also investigated. Based on the analysis of obtained results, the mechanism of biofilm formation of the MSSA strain was different from that of the MRSA strain analysed by Qin et al. [85]. The difference is due to the lack of the role of an accessory gene regulator (*agr*) in the MSSA strain. These findings also suggest that biofilms of the MSSA strain may be more resistant to antibiotics than biofilms of the MRSA strain that has a fully functional *agr*.

Studies determining the effect of UA and UA combined with ciprofloxacin on the process of biofilm formation and eradication were also conducted by Wojnicz et al. [49]. The researchers used polystyrene microtiter plates and silicone urological catheters as adhesive surfaces. The reference *E. coli* strain CFT073 (ATCC 700928) and 10 uropathogenic clinical *E. coli* strains with genes encoding proteins that are important in biofilm formation were used in those studies. On titre plates, both ursolic acid and its combination with ciprofloxacin showed anti-biofilm activity, especially in older biofilms. There was a decrease in both the amount of produced biofilm mass and the number of viable bacteria. Unfortunately, UA used alone had a weak effect on the eradication of biofilm from urological catheters. Statistically significant eradication of the biofilm mass was obtained only after treatment with a mixture of UA and ciprofloxacin.

The subject of a study by Feuillolay et al. [107] was *P. acnes*, an opportunistic strain of Gram-positive bacteria that is resistant to many tetracyclines and macrolide antibiotics. *P. acnes* grows as a biofilm on biomedical materials (implants) and in hair follicles of the skin, causing acne vulgaris. *Myrtus communis* leaf extract, in which UA content was quantified by HPLC and it was 20%, showed significant antibacterial activity against *P. acnes* strains that are insusceptible to erythromycin and clindamycin, growing both in suspended matter and forming a biofilm. The analysed extract, at concentrations of 10–1000 mg/L, inhibited biofilm formation and reduced the structured 48-h biofilm of *P. acnes*. The authors also determined the antibacterial properties of the extract (10 mg/L) combined with erythromycin (1000 mg/L) or clindamycin (500 mg/L). In those combinations, the analysed extract restored susceptibility of *P. acnes* strains to both erythromycin and clindamycin.

According to Chung et al. [94], when added to standard dental material (3 mg UA per 1 mL material), UA had the ability to inhibit biofilm formation by *S. mutans* UA159 on the tooth surface. 

Ray et al. [108] analysed the effect of UA (30 mg/L) on biofilm mass synthesis by a clinical strain of *S. marcescens*. These bacilli may be the cause of catheter-related UTIs. UA was found to inhibit biofilm formation by the analysed bacterial strain.

Other Wojnicz et al. studies [46] investigated the effects of UA on biofilm production and survival of ten clinical *E. faecalis* strains. Although UA at a concentration of 0.75 × MIC did not exhibit any significant inhibitory effect on biofilm mass synthesis, it significantly reduced the survival of cocci at all stages of 10-day biofilm development.

Jyothi et al. [109] investigated the antibiofilm activity of UA against 50 *S. aureus* strains with *icaD* adhesion gene involved in biofilm production. Inhibition of biofilm formation was observed in 40 isolates of the analysed strains, and it was 48.6% for UA applied at a concentration of 30 mg/L and 71.5% when the UA concentration was 60 mg/L, respectively.

Silva et al. [110] investigated the activity of UA (5, 25, and 100 μM) isolated from an apple peel (*Malus domestica*) against Gram-positive bacteria: *E. faecalis* ATCC 29212, *S. aureus* ATCC 25904, and *S. epidermidis* ATCC 35984. Only UA at a concentration of 100 μM showed antimicrobial activity against planktonic cells and it inhibited biofilm synthesis by all bacterial strains under analysis.

As has been demonstrated by Liu et al. [13], UA decreased the viability of *S. mutans* and the structural integrity of its biofilms by interacting with the catalytic centre of glucosyltransferases, the key enzymes required in EPS synthesis. Lyu et al. [14] found that UA reduced the formation of multi-species biofilms (*S. mutans*, *S. sanguinis*, and *S. gordonii*) by inhibiting the expression of *gft* genes and in consequence leading to inhibition of EPS formation.

## 8. Conclusions

The studies published so far show that antibacterial activity of AA and UA is related to changes in the structure and functioning of the bacterial cell structures (cell membrane, adhesins), cell morphology, expression of genes encoding virulence factors such as P fimbriae, curli fimbriae, and hydrophobicity. Both pentacyclic triterpenes can affect the adhesion of bacteria to host cells and the process of biofilm formation, but the exact molecular mechanisms of this activity are still not fully explained. Therefore, our article also presents and summarises suggested by researchers’ mechanisms of the biological action of these substances against microorganisms.

The article describes how to improve the poor availability of acids in vivo that limits their clinical application. Structural modifications of these substances have been conducted in recent years to improve their biological activity and bioavailability, such as designing and synthesising novel derivatives, improvement of its water solubility, encapsulation in carries (i.e., nanostructures).

In our article, we also intend to highlight that in the perspective of further research, the existence of synergistic effects of AA and UA with antibiotics (i.e., β-lactams, tetracyclines, fluoroquinolones, aminoglycosides) should be taken into account. It is even more necessary to define the rules for antimicrobial activity of both acids validation and its conversion of in vitro potency into in vivo therapeutic activity. Then, AA and UA could serve as supplements to standard pharmacotherapy.

## Figures and Tables

**Figure 1 biomolecules-12-00098-f001:**
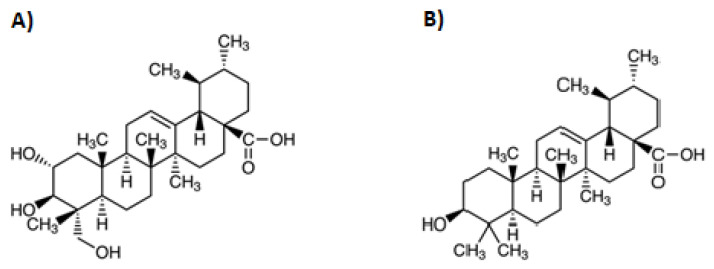
Chemical formulas of asiatic (**A**) and ursolic (**B**) acids.

**Figure 2 biomolecules-12-00098-f002:**
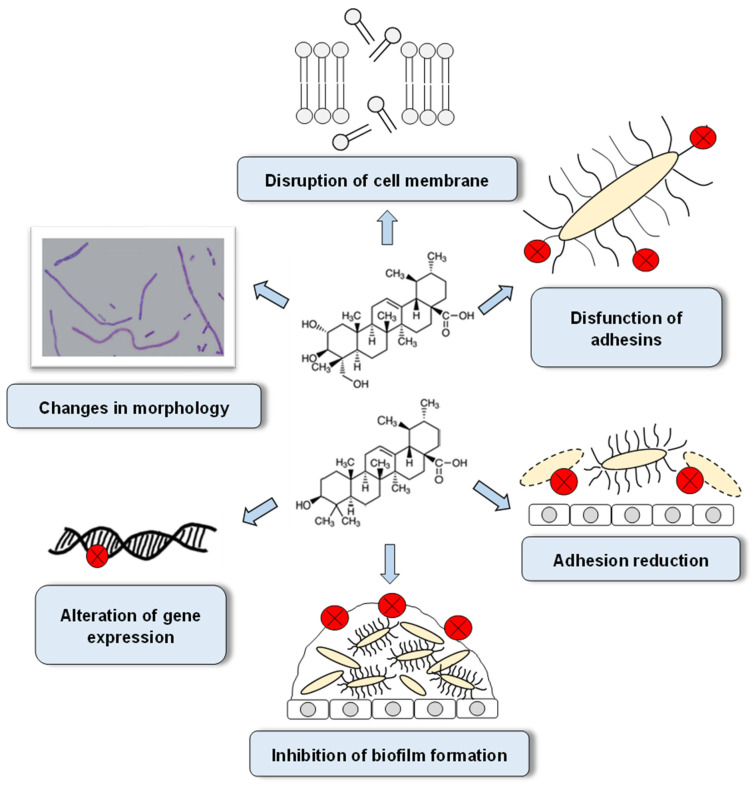
The various antibacterial targets of asiatic and ursolic acids.

**Figure 3 biomolecules-12-00098-f003:**
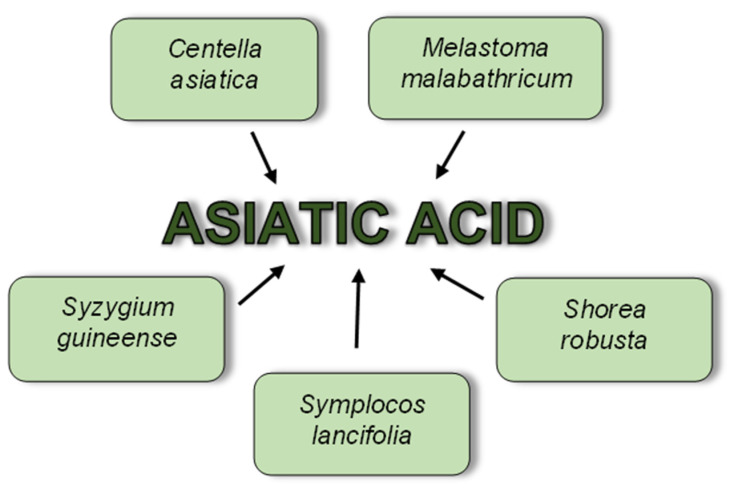
Plant sources of asiatic acid used in antibacterial experiments; green boxes—angiosperms.

**Figure 4 biomolecules-12-00098-f004:**
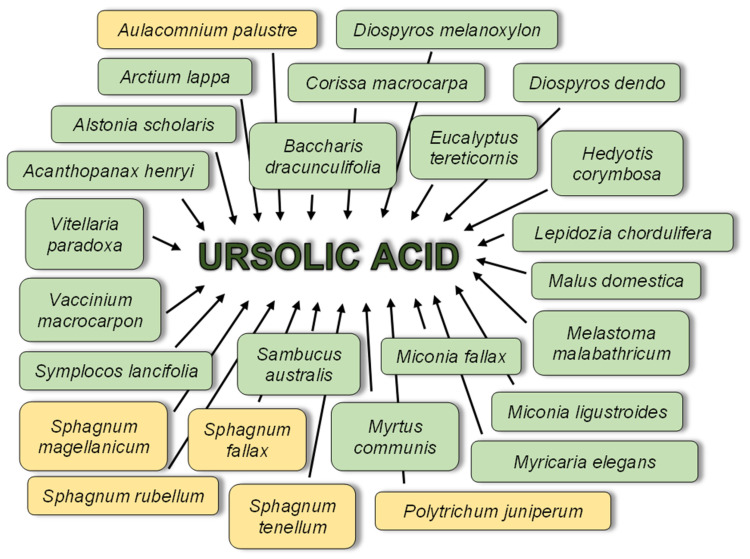
Plant sources of ursolic acid used in antibacterial experiments; green boxes—angiosperms; yellow boxes—bryophytes.

## Data Availability

Not applicable.

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
