# Peer review of "Anti-Planktonic and Anti-Biofilm Properties of Pentacyclic Triterpenes—Asiatic Acid and Ursolic Acid as Promising Antibacterial Future Pharmaceuticals"

_biomolecules, 2022, doi:10.3390/biom12010098_

Round 1
Reviewer 1 Report
The manuscript of Sycz et al. reviews the current knowledge about the anti-planktonic and anti-biofilm bacterial properties of the pentacyclic triterpenes asiatic acid (AA) and ursolic acid (UA).
Comments:
- Abstract, line 18: I suggest removing ”suggested by researchers”.
- Long phrases with new information without any reference. Please add appropriate citations to: Page 1, lines 34 to 42; Page 3, lines 94 to 96; Page 3, lines 97-100; Page 4., lines 150 to 166; Page 6, lines 271 to 279;
- Page 4, lines 153-154: Please rephrase.
- Confirm if all species name are in italic. Example page 6, line 258 is not in italic.
- Figure 3 and 4: In my opinion the arrows should be in the opposite direction, because these plants are the source of the AA or UA and the image at first impression seems that AA or UA is doing something on these plants.
- Table 1 and 3, Source column: Some AA or UA have as source plants and other were purchased (???). Please clarify the difference, if they are not from plant origin, or if the origin is unknown, or the different works use the same AA and the origin is not described?
- Different sources of AA or UA give different results, these is not discussed in the manuscript. For example, AA from C. asiatica is always more active than M. malabathticum in table 2. Other examples, different purchased AA give different mic results, what are the differences in these studies? The same for the different sources of UA.
- It is not also discussed why AA extracted from C. asiatica give different results in study ref. 52 and ref. 51. For example the extraction method is the same?
- In Page 9, line 355: Again the results of table 1 are described. I suggest that these part of the text will be included before the descriptions of the results of Table 2.
- Page 10, lines 365-367: These information is not included in table 1.
- Page 15, line 493-495: These observation from the results presented in tables 1 to 4 is true for UA, however for AA is not so clear. In table 2, AA had active or very active antibacterial activity against several gram-negative bacteria. Please confirm.
Author Response
We greatly appreciate your valuable comments and time spent reviewing our manuscript. We are convinced that this has significantly increased its quality.
Comments and Suggestions for Authors
Reviewer comment: Abstract, line 18: I suggest removing ”suggested by researchers”.
Authors’ response: the fragment of the sentence has been removed
Reviewer comment: Long phrases with new information without any reference. Please add appropriate citations to: Page 1, lines 34 to 42; Page 3, lines 94 to 96; Page 3, lines 97-100; Page 4., lines 150 to 166; Page 6, lines 271 to 279;
Authors’ response: required references have been added
Reviewer comment: Page 4, lines 153-154: Please rephrase.
Authors’ response: the sentence has been rephrased
Reviewer comment: Confirm if all species name are in italic. Example page 6, line 258 is not in italic.
Authors’ response: corrected - species names are in italic
Reviewer comment: Figure 3 and 4: In my opinion the arrows should be in the opposite direction, because these plants are the source of the AA or UA and the image at first impression seems that AA or UA is doing something on these plants.
Authors’ response: thank you for this suggestion, you are right; the direction of the arrows has been changed and now the Figures look much better.
Reviewer comment: Table 1 and 3, Source column: Some AA or UA have as source plants and other were purchased (???). Please clarify the difference, if they are not from plant origin, or if the origin is unknown, or the different works use the same AA and the origin is not described?
Authors’ response: thank you for this suggestion, we clarified the difference and added the origin of the purchased acids (plant origin or synthetic origin) in the manuscript
Reviewer comment: Different sources of AA or UA give different results, these is not discussed in the manuscript. For example, AA from C. asiatica is always more active than M. malabathticum in table 2. Other examples, different purchased AA give different mic results, what are the differences in these studies? The same for the different sources of UA.
Authors’ response: When analyzing the antibacterial activity of AA or UA, it can be noticed that their antimicrobial activity (MIC/inhibition zone) may be different for the same bacterial species. However, there are several different aspects to consider. First, there are many different bacterial strains within a species, all from different origins (biological material from which it was isolated, e.g. urine, faeces, blood, etc.). Secondly, the acids used in separate studies often have different origins, i.e. they are isolated from different plant species. Moreover, the methods of extracting these metabolites often vary.
Different MIC values obtained after using UA purchased in the same company result from the difference in the origin of bacterial strains - an example is the study of Fontanay et al. [72] who used Pseudomonas aeruginosa ATCC 27853 (blood isolate; MIC ≥256 mg/L) and Kurek et al. [75] tested P. aeruginosa PCM 499 (unknown isolation source; MIC = 22.5 mg/L). On the other hand, Fontanay et al. [72] and Oloyede et al. [84] used the same P. aeruginosa strain (ATCC 27853) and the purchased synthetic UA, however, obtained different MIC values. Unfortunately, the authors did not describe either the way of the stock solution preparation or the type of solvent. Therefore, such results are difficult to discuss.
The other example, are the differences obtained by Wong et al. [67] and Ashella et al. [66] who teste the same species of bacterial strains (K. pneumoniae and S. aureus ). Probably, these differences can be associated with the method of extraction and isolation of AA from plant material. Ashella et al. state that they obtained AA from C. asiatica, but in the article, there is no method of AA extraction described. Additionally, both research groups used the same bacterial species, but the strains had different origins. Wong et al. [67] used S. aureus ATCC 25923 and Ashella et al. [66] from Microbial Type Culture Collection (MTCC), Chandigarh, Punjab, India.
Moreover, the differences in MIC values for purchased AA are due to the different origin of the acid – e.g. in Table 1 AA purchased from Sigma-Aldrich Chemicals Inc. was synthetic (assigned as A) or isolated from C. asiatica (assigned as B).
Reviewer comment: It is not also discussed why AA extracted from C. asiatica give different results in study ref. 52 and ref. 51. For example the extraction method is the same?
Authors’ response: When analyzing the studies of Ashella et al. [66] – the authors stay that they isolated AA from C. asiatica but unfortunately they do not describe the method of extraction and isolation of AA. On the other hand, Norzaharaini et al. [68] inform that they used purchased AA, but do not let the readers know whether it is synthetic one or extracted from C. asiatica. Therefore, such results are difficult to discuss.
Reviewer comment: In Page 9, line 355: Again the results of table 1 are described. I suggest that these part of the text will be included before the descriptions of the results of Table 2.
Authors’ response: Subsection 6.1. has been reorganized. Study descriptions for MIC values have been grouped together and are provided under Table 1. Similarly, descriptions of zones of growth inhibition are summarized below Table 2.
Reviewer comment: Page 10, lines 365-367: These information is not included in table 1.
Authors’ response: We analyzed once again the study of Norzaharaini et al. [68]. The information we omitted earlier is now presented in Table 2.
Reviewer comment: Page 15, line 493-495: These observation from the results presented in tables 1 to 4 is true for UA, however for AA is not so clear. In table 2, AA had active or very active antibacterial activity against several gram-negative bacteria. Please confirm.
Authors’ response: thank you, you are right. We reorganized the description of our observation in the text of the manuscript
Reviewer 2 Report
This article described the research progress of anti-planktonic and anti-biofilm properties of pentacyclic triterpenes: asiatic acid, ursolic acid and their derivatives which would be promising antibacterial pharmaceuticals in the future has made contribution to predict the development and research trend of antibacterial drugs or seek new research breakthroughs to bacterial infectious diseases, and it merits publication in Biomolecules. But there are still have some problems need to be improved before it can be accepted for publication.
1) The summary of bacterial biofilm in the introduction section is too broad, current research progress of anti-planktonic and anti-biofilm should be noted in the this part;
2) The references of this article is not compete, please check the research topic in difference data bases so that you can obtained more important information which is good for your research;
3) Some formats of this article is not completely correct. For example, the unit of milliliter in this manuscript is not standardized; and line 324 : 1024 µl/ml, a wrong unit;
4) The manuscript still exits some English language problems including grammar, spelling, and sentence structure, please check carefully with someone who expertise in technical English editing. For example, line383, 551: the value should use the plural form of the noun;
Please check other mistakes not mentioned here.

Author Response
We greatly appreciate your valuable comments and time spent reviewing our manuscript. We are convinced that this has significantly increased its quality.
Reviewer comment: 1) The summary of bacterial biofilm in the introduction section is too broad, current research progress of anti-planktonic and anti-biofilm should be noted in the this part;
Authors’ response: The description of biofilm in the Introduction has been shorten. Current research conducting the anti-planktonic and anti-biofilm activities of pentacyclic triterpenes have been added.
Reviewer comment: 2) The references of this article is not complete, please check the research topic in difference data bases so that you can obtained more important information which is good for your research;
Authors’ response: The new literature positions (from 2021) describing the research topic have been added to the manuscript.
Reviewer comment: 3) Some formats of this article is not completely correct. For example, the unit of milliliter in this manuscript is not standardized; and line 324 : 1024 µl/ml, a wrong unit;
Authors’ response: thank you, should be µg instead of µl; we also corrected all the units according to SI guidelines.
Reviewer comment: 4) The manuscript still exits some English language problems including grammar, spelling, and sentence structure, please check carefully with someone who expertise in technical English editing. For example, line383, 551: the value should use the plural form of the noun;
Authors’ response: English language has been checked and improved.
Reviewer comment: Please check other mistakes not mentioned here.
Authors’ response: We hope that all the other mistakes have been corrected.
Reviewer 3 Report
The authors describe anti-planktonic and anti-biofilm properties of pentacyclic triterpenes, asiatic acid and ursolic acid. The manuscript is well organized. However, it seems that the overall quality of the picture is not good. I think that the quality of the figure in the review paper is quite important.
Author Response
We greatly appreciate your valuable comment and time spent reviewing our manuscript. We are convinced that this has significantly increased its quality.
Reviewer comment: The authors describe anti-planktonic and anti-biofilm properties of pentacyclic triterpenes, asiatic acid and ursolic acid. The manuscript is well organized.
However, it seems that the overall quality of the picture is not good. I think that the quality of the figure in the review paper is quite important.
Authors’ response: The quality of the Figures has been improved.
Reviewer 4 Report
The Review: Anti-Planktonic and Anti-Biofilm Properties of Pentacyclic Triterpenes - Asian Acid and Ursolic Acid as Promising Antibacterials
future pharmaceuticals, provides the reader with a broader understanding of the activity of these acids, isolated from plants, against pathogenic bacteria. The review is well written and makes a significant contribution to the topic. The only deficiency I find, and which I invite the authors to fill, is the absolute contempt for the presence and activity of these substances even in bryophytes. Just a few accounts in the introduction would make it complete.
Author Response
We greatly appreciate your valuable comment and time spent reviewing our manuscript. We are convinced that this has significantly increased its quality.
Reviewer comment: The only deficiency I find, and which I invite the authors to fill, is the absolute contempt for the presence and activity of these substances even in bryophytes. Just a few accounts in the introduction would make it complete.
Authors’ response: Thank you for this suggestion. We checked the research topic in different databases and added literature positions describing the presence of ursolic acid in several species belonging to bryophytes and also ferns.
Round 2
Reviewer 1 Report
The authors answered to the majority of the comments made, so the present manuscript version is aceptable for publication.
The only suggestion that I have is that altough is dificult to discuss the "different sources of AA or UA give different results". The answer given by the authors to this comment could be used as base for an brief discussion to these results.